# Autism-related KLHL17 and SYNPO act in concert to control activity-dependent dendritic spine enlargement and the spine apparatus

Hsiao-Tang Hu[1], Yung-Jui Lin[1], Ueh-Ting Tim Wang[2,3], Sue-Ping Lee[1], Yae-Huei Liou[1], Bi-Chang Chen[3], Yi-Ping Hsueh[1]*

1 Institute of Molecular Biology, Academia Sinica, Taipei, Taiwan, 2 Affiliated Senior High School of National Taiwan Normal University, Taipei, Taiwan, 3 Research Center for Applied Sciences, Academia Sinica, Taipei, Taiwan

* yph@gate.sinica.edu.tw

**Data Availability Statement:** All relevant data are within the paper and its Supporting Information files. The file of S1 Data contains the numerical value data of all figures. S2 Data contains all

## Abstract

Dendritic spines, the tiny and actin-rich protrusions emerging from dendrites, are the subcellular locations of excitatory synapses in the mammalian brain that control synaptic activity and plasticity. Dendritic spines contain a specialized form of endoplasmic reticulum (ER), i.e., the spine apparatus, required for local calcium signaling and that is involved in regulating dendritic spine enlargement and synaptic plasticity. Many autism-linked genes have been shown to play critical roles in synaptic formation and plasticity. Among them, KLHL17 is known to control dendritic spine enlargement during development. As a brain-specific disease-associated gene, *KLHL17* is expected to play a critical role in the brain, but it has not yet been well characterized. In this study, we report that KLHL17 expression in mice is strongly regulated by neuronal activity and KLHL17 modulates the synaptic distribution of synaptopodin (SYNPO), a marker of the spine apparatus. Both KLHL17 and SYNPO are F-actin-binding proteins linked to autism. SYNPO is known to maintain the structure of the spine apparatus in mature spines and contributes to synaptic plasticity. Our super-resolution imaging using expansion microscopy demonstrates that SYNPO is indeed embedded into the ER network of dendritic spines and that KLHL17 is closely adjacent to the ER/SYNPO complex. Using mouse genetic models, we further show that *Klhl17* haploinsufficiency and knockout result in fewer dendritic spines containing ER clusters and an alteration of calcium events at dendritic spines. Accordingly, activity-dependent dendritic spine enlargement and neuronal activation (reflected by extracellular signal-regulated kinase (ERK) phosphorylation and C-FOS expression) are impaired. In addition, we show that the effect of disrupting the KLHL17 and SYNPO association is similar to the results of *Klhl17* haploinsufficiency and knockout, further strengthening the evidence that KLHL17 and SYNPO act together to regulate synaptic plasticity. In conclusion, our findings unravel a role for KLHL17 in controlling synaptic plasticity via its regulation of SYNPO and synaptic ER clustering and imply that impaired synaptic plasticity contributes to the etiology of KLHL17-related disorders.

statistical results. S1 Raw Images contains uncropped blots.

**Funding:** This work was supported by grants from Academia Sinica (https://www.sinica.edu.tw, AS-IA-111-L01 and AS-TP-110-L10 to Y.-P.H.), and the National Science and Technology Council (https://www.nstc.gov.tw/?, NSTC 108-2311-B-001-008-MY3 to Y.-P.H.). The funders had no role in study design, data collection and analysis, the decision to publish or the preparation of the manuscript.

**Competing interests:** The authors have declared that no competing interests exist.

**Abbreviations:** AF, actinfilin; AMPAR, α-amino-3-hydroxy-5-methyl-4-isoxazolepropionic acid receptor; AP5, 2-amino-5-phosphonopentanoic acid; ASD, autism spectrum disorder; BTB, Bric-a-brac/Tramtrack/Broad; CHX, cyclohexamide; DIV, day in vitro; ER, endoplasmic reticulum; ERK, extracellular signal-regulated kinase; KLHL17, Kelch-like protein 17; mTOR, mammalian target of rapamycin; NBQX, 2,3-dihydroxy-6-nitro-7-sulfamoyl benzo(f)quinoxaline; NMDAR, N-methyl-D-aspartate receptor; RFU, relative fluorescence unit; RT-PCR, real time-PCR; SYNPO, synaptopodin; TTX, tetrodotoxin.

## Introduction

Autism spectrum disorders (ASDs) are highly prevalent neuropsychiatric disorders characterized by impaired social and communication behaviors, abnormal sensations, and stereotyped activities [1]. Human genetic studies have identified hundreds of genes associated with ASD (https://gene.sfari.org/database/human-gene/). Many of these disease-risk genes are directly or indirectly involved in synaptic formation, signaling, and plasticity [2–5]. Accordingly, it has been hypothesized that perturbation of those ASD-linked genes may increase or decrease synaptic number and/or strength, consequently promoting abnormal neuronal connectivity in the brain [5–10]. Thus, synaptopathy is highly relevant to ASD etiology.

Among the various ASD-associated genes, *Kelch-like protein 17* (*KLHL17*), also known as actinfilin [11,12], has been shown to contribute to dendritic spine enlargement during development [13]. *Klhl17*$^{+/−}$ mice exhibit social deficits and hyperactive locomotion [13], echoing genetic evidence from patients that *Klhl17* deficiency is associated with ASD [2,14]. Since KLHL17 controls dendritic spine enlargement, this protein may serve as a model to explore how morphological plasticity is relevant to ASD etiology.

As a member of the Kelch-like protein family, KLHL17 contains a Bric-a-brac/Tramtrack/Broad complex (BTB) domain at its N-terminal region and 6 Kelch domains at the C-terminal half [11,15–17]. The BTB domain has been shown to mediate dimerization [18] and interaction with CUL3 ubiquitin E3 ligase [15,17]. The Kelch domains also act as a protein–protein interacting domain to recognize CUL3 substrates [15,16,19]. The interaction between KLHL17 and F-actin cytoskeletons is also mediated by the Kelch domains [11,13]. Knockdown or knockout of *Klhl17* or disruption of the KLHL17 and F-actin interaction in mouse neurons impairs dendritic spine targeting of F-actin and disrupts dendritic spine enlargement during the developmental process [13]. Given that KLHL17 protein is specifically expressed in the brain [11], it has been suggested that KLHL17 exerts neuron-specific functions to control the morphology of excitatory synapses, i.e., neuron-specific and F-actin-enriched subcellular structures [13].

Nevertheless, how *Klhl17* deficiency affects neuronal responses and functions remains elusive. We reported here that neuronal activation increases KLHL17 protein levels via glutamate receptor and protein synthesis. Using cultured mouse neurons, we demonstrate that KLHL17 is a critical factor involved in controlling activity-dependent dendritic spine enlargement. KLHL17 associates with synaptopodin (SYNPO), a marker of the spine apparatus (i.e., endoplasmic reticulum (ER) located at dendritic spines) [20,21]. SYNPO also associates with F-actin via its interaction with actinin [22]. We show that KLHL17 and SYNPO work together to control the synaptic clustering and distribution of the spine apparatus and calcium dynamics at dendritic spines. Ultimately, these functions influence neuronal activation, as reflected by extracellular signal-regulated kinase (ERK) phosphorylation and C-FOS expression. Given that SYNPO is also linked to ASD [23], our study strengthens the relevance of ASD etiology to synaptic plasticity and the calcium dynamics controlled by the spine apparatus.

## Results

### KLHL17 protein levels are regulated in development- and activity-dependent manners

When we examined KLHL17 expression in mouse cortical and hippocampal mixed cultures, we observed that its protein levels gradually increased as the cultures matured (**Fig 1A and 1B**, upper). In mouse brains, KLHL17 protein levels also gradually increased from postnatal day 1 to 21 (**Fig 1A and 1B**, lower). However, in contrast to the increased protein levels, quantitative PCR revealed that levels of *Klhl17* mRNAs were actually reduced in mouse brains and even

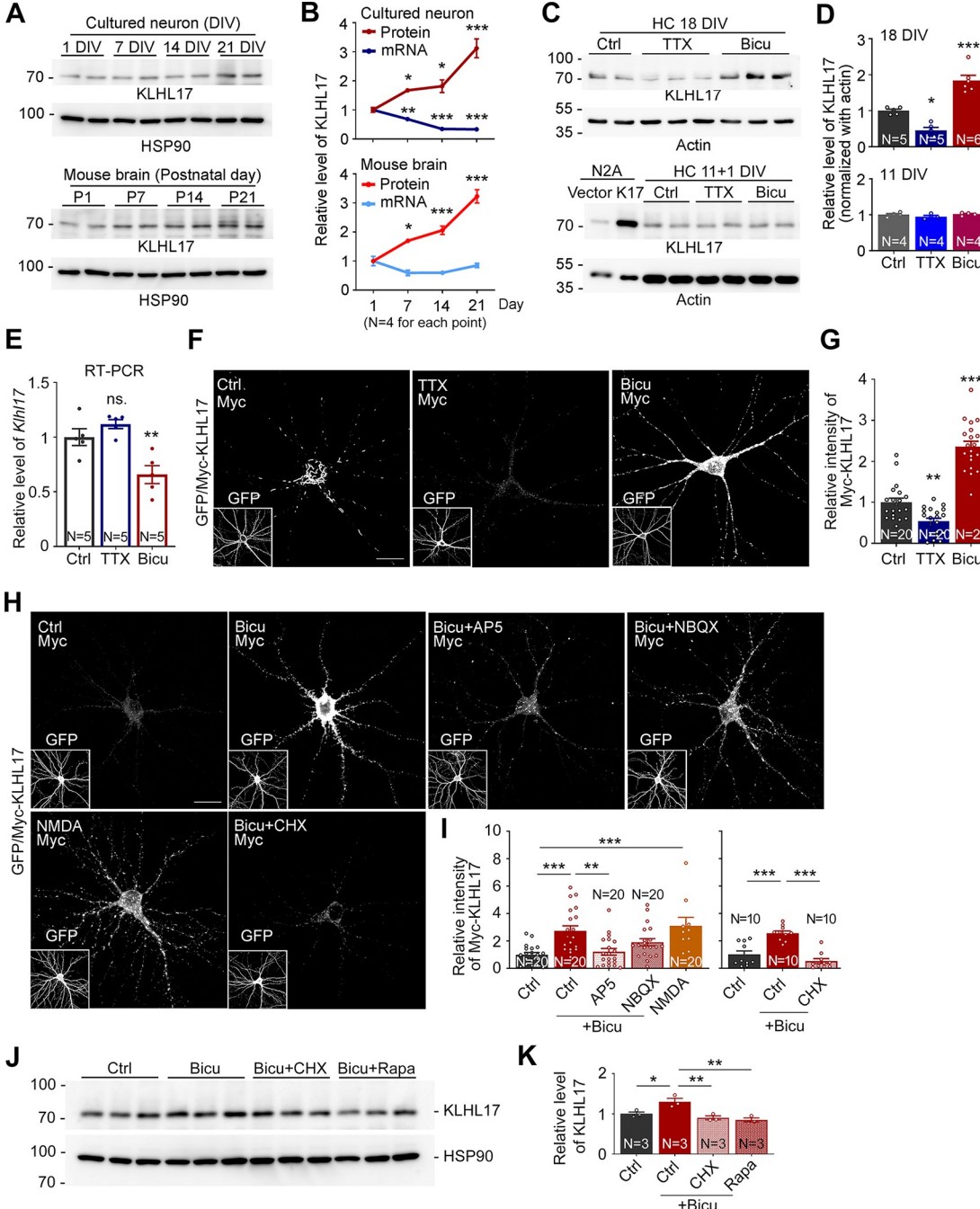

**Fig 1. Increased neuronal activity up-regulates KLHL17 protein levels.** (**A**, **B**) Protein levels, but not RNA levels, of KLHL17 are increased as neurons mature. Upper panel: Total cell lysate prepared from cultured neurons at different time points, i.e., 1, 7, 14, and 21 DIV. Lower panel: A total of 10 μg of tissue lysates isolated from whole brain of mice of different ages, i.e., postnatal days (P) 1, 7, 14, and 21. Quantifications of relative protein and RNA levels of the *Klhl17* gene are shown in (**B**). (**C**, **D**) Protein levels of KLHL17 are increased by neuronal activation. Mature cultured neurons (18 DIV, upper) and immature cultured neurons (11 DIV, lower) were treated with TTX (1 μM), bicuculline (Bicu, 40 μM), and vehicle control (Ctrl) for 6 and 24 h, respectively. Neuro-2A (N2A) cell lysate transfected with KLHL17 acted a positive control. K17 and KLHL17 are interchangeable in the figure. Quantification is shown in (**D**). (**E**) Endogenous RNA levels of *Klhl17* are reduced by bicuculline treatment. (**F**–**I**) Exogenous KLHL17 is sensitive to neuronal activity via NMDAR signaling and protein synthesis. Cultured neurons were transfected with Myc-KLHL17 and GFP at 12 DIV and then subjected to various treatments for 6 h at 18 DIV as indicated. NMDA, 10 μM; AP5, 100 μM; NBQX, 100 μM; CHX, 10 μM. (**F**, **H**) Representative images. GFP images shown in insets indicate transfected cells. (**G**, **I**) Quantification of exogenous Myc-KLHL17 signals. (**J**, **K**) The mTOR pathway and translation regulate KLHL17 protein levels. The effects of CHX (10 μM) or Rapamycin (10 nM) treatment were assessed. (**J**) Total cell lysates were analyzed by

immunoblotting. (**K**) Quantification of relative protein levels of KLHL17. All immunoblots were performed using antibodies recognizing KLHL17 and internal control (HSP90 or actin, as indicated). The cultures were randomly assigned to treatments. For immunoblotting, each lane represents an independent sample. The sample sizes (N) of independent preparations or examined neurons are indicated in the panels. The data represent mean ± SEM. Individual data points are also shown. * $P < 0.05$; ** $P < 0.01$; *** $P < 0.001$; ns, not significant; one-way ANOVA. Scale bars: (**F**, **H**) 20 μm. The numerical value data and statistical results are available in S1 and S2 Data, respectively. DIV, day in vitro; KLHL17, Kelch-like protein 17; mTOR, mammalian target of rapamycin; NBQX, 2,3-dihydroxy-6-nitro-7-sulfamoyl benzo(f)quinoxaline; NMDAR, N-methyl-D-aspartate receptor.

more so in cultured neurons (**Fig 1B**), indicating that KLHL17 protein levels are posttranscriptionally up-regulated during development.

Given that increased neuronal activity is a key feature of maturing neurons, we investigated if the increased KLHL17 protein levels are relevant to enhanced neuronal activity. Under our culture conditions, cortical and hippocampal mixed cultures become fully mature at approximately 18 days in vitro (DIV) [24,25]. We treated the mature cultures with the sodium channel blocker tetrodotoxin (TTX) to inhibit neurotransmission and with the GABA_A receptor antagonist bicuculline to enhance neuronal activity. Bicuculline-enhanced neuronal activity indeed increased KLHL17 protein levels, whereas limiting neuronal activity by means of TTX treatment reduced them (**Fig 1C and 1D**, upper). These alterations were not observed for immature cultures at 11 DIV (**Fig 1C and 1D**, lower), supporting that neuronal maturation is involved in controlling KLHL17 expression. In contrast to the increased protein levels upon neuronal activation, RNA levels of *Klhl17* were reduced by bicuculline but remained unaltered upon TTX treatment (**Fig 1E**), revealing that neuronal activation exerts opposing control on the protein and RNA levels of *Klhl17*.

Like endogenous KLHL17, expression levels of exogenous KLHL17 in mature neurons also responded in the same fashion to TTX and bicuculline treatments (**Fig 1F and 1G**), further supporting that KLHL17 proteins are controlled by neuronal activity.

Taken together, these results show that neuronal activation up-regulates KLHL17 protein expression, even though RNA levels of *Klhl17* are not increased or are even reduced. Thus, a complex regulatory mechanism is involved in controlling the RNA and protein levels of *Klhl17* in opposing directions. Our findings also imply a critical role for synaptic stimulation in controlling KLHL17 expression.

## NMDAR signaling and protein synthesis control KLHL17 proteins levels

Next, we investigated if glutamate receptors, the major excitatory neurotransmitter receptors in mammalian brains, are involved in regulating KLHL17 protein levels. AP5, an N-methyl-D-aspartate receptor (NMDAR) blocker, but not NBQX, an antagonist of the AMPA receptor, completely prevented the increase in KLHL17 protein levels induced by bicuculline (**Fig 1H and 1I**). Consistently, NMDA treatment also increased KLHL17 protein levels (**Fig 1H and 1I**), supporting that the NMDAR pathway is critical for controlling protein levels of KLHL17.

We then applied cycloheximide, an inhibitor of translational elongation, and rapamycin, a blocker of the mammalian target of rapamycin (mTOR) pathway for translation, to bicuculline-treated cultures. Inhibiting protein synthesis in this way blocked the ability of bicuculline to enhance both exogenous and endogenous KLHL17 protein levels (**Fig 1H and 1K**).

Thus, neuronal activity via NMDAR signaling and protein synthesis tightly controls KLHL17 protein expression.

## KLHL17 is required for activity-dependent enlargement of dendritic spines and neuronal activation

Given that KLHL17 is required for the enlargement of dendritic spine heads during development [13] and that neuronal activation up-regulates KLHL17 expression (**Fig 1**), we speculated that

KLHL17 is also involved in the neuronal activity-dependent structural plasticity of dendritic spines. To investigate that possibility, we subjected neurons to bicuculline treatment for 15 min to stimulate them, followed by recovery for 30 min (**Fig 2A**). First, we tested the response of *Klhl17* knockdown hippocampal neurons, generated by transfecting a previously established *Klhl17*-miR construct, i.e., AF-miR [13], into cultured neurons. Consistent with a previous report [13], expression of *Klhl17*-miR narrowed dendritic spine width but did not affect the density or length of dendritic spines compared with a scrambled control group (Ctrl-miR) (**Fig 2B–2E**). After activation, neurons transfected with Ctrl-miR displayed noticeably broader spine heads, but the density and length of their dendritic spines remained unaltered (**Fig 2B–2E**), reflecting activity-dependent dendritic spine enlargement. However, for the *Klhl17* knockdown neurons, the same bicuculline treatment did not increase, but actually slightly decreased, the width of dendritic spine heads (**Fig 2B and 2D**), indicating that KLHL17 is required for activity-dependent dendritic spine enlargement. The effect of KLHL17 on dendritic spine head width was specific because the density and length of dendritic spines were not altered by *Klhl17* knockdown.

To further validate this finding, we subjected both *Klhl17*$^{+/-}$ and *Klhl17*$^{-/-}$ neurons to bicuculline treatment. Similar to our results for *Klhl17* knockdown neurons, we observed that *Klhl17*$^{+/-}$ and *Klhl17*$^{-/-}$ neurons displayed narrow spine heads and did not respond to bicuculline treatment (**Fig 2F–2I**). Note that deletion of one of the *Klhl17* alleles proved sufficient to impair activity-dependent dendritic spine enlargement (**Fig 2F–2I**). Deletion of both alleles did not further exacerbate the phenotype. These results reveal that *Klhl17* haploinsufficiency impairs activity-dependent dendritic spine enlargement, consistent with genetic features of patients in which only 1 allele of the *Klhl17* gene is mutated or deleted [2,14,26].

Since spine enlargement is a key feature of the synaptic response [27,28], failure of *Klhl17*-deficient neurons to enlarge their dendritic spines upon bicuculline treatment implies the possibility of impaired neuronal activation. To explore that possibility, we investigated neuronal activity using 2 indicators, i.e., ERK phosphorylation and C-FOS expression (**Fig 3A**). Basal levels of ERK phosphorylation and C-FOS expression were very low under our culture conditions (**S1 Fig**). Compared with wild-type neurons, both *Klhl17*$^{+/-}$ and *Klhl17*$^{-/-}$ neurons exhibited lower levels of ERK phosphorylation following bicuculline treatment for 15 min (**Figs 3B–3E and S1**). Moreover, 2 h after bicuculline treatment, we found that much fewer *Klhl17*$^{+/-}$ and *Klhl17*$^{-/-}$ neurons expressed C-FOS, an immediate early marker of neuronal activation, compared to wild-type neurons (**Figs 3F, 3G and S1**). These results indicate a reduction in neuronal activation attributable to *Klhl17* deficiency.

Thus, *Klhl17* deficiency results in impaired spine remodeling and defective neuronal activation upon bicuculline treatment.

## KLHL17 regulates calcium dynamics

Activity-dependent spine enlargement is deemed relevant to spine apparatus-dependent calcium signaling [29–32]. We speculated that KLHL17 regulates spine apparatus-dependent calcium influx into the cytosol. To test that possibility, we first investigated if *Klhl17* deficiency alters calcium dynamics in neurons. Cortical and hippocampal mixed cultures were transfected with GCaMP6s at 12 DIV and then we monitored relative changes in calcium concentrations in the cytoplasm of neurons based on GCaMP6s fluorescence signals at 18 DIV (**Fig 4A**). More specifically, we analyzed the frequency and amplitude of calcium events and the interval between events at dendritic spines. We found that the frequency of calcium events was reduced and the interval between calcium events was increased in *Klhl17*$^{+/-}$ neurons compared with wild-type neurons (**Fig 4B–4D**). However, the amplitude of calcium events was enhanced by *Klhl17* deficiency (**Fig 4E**).

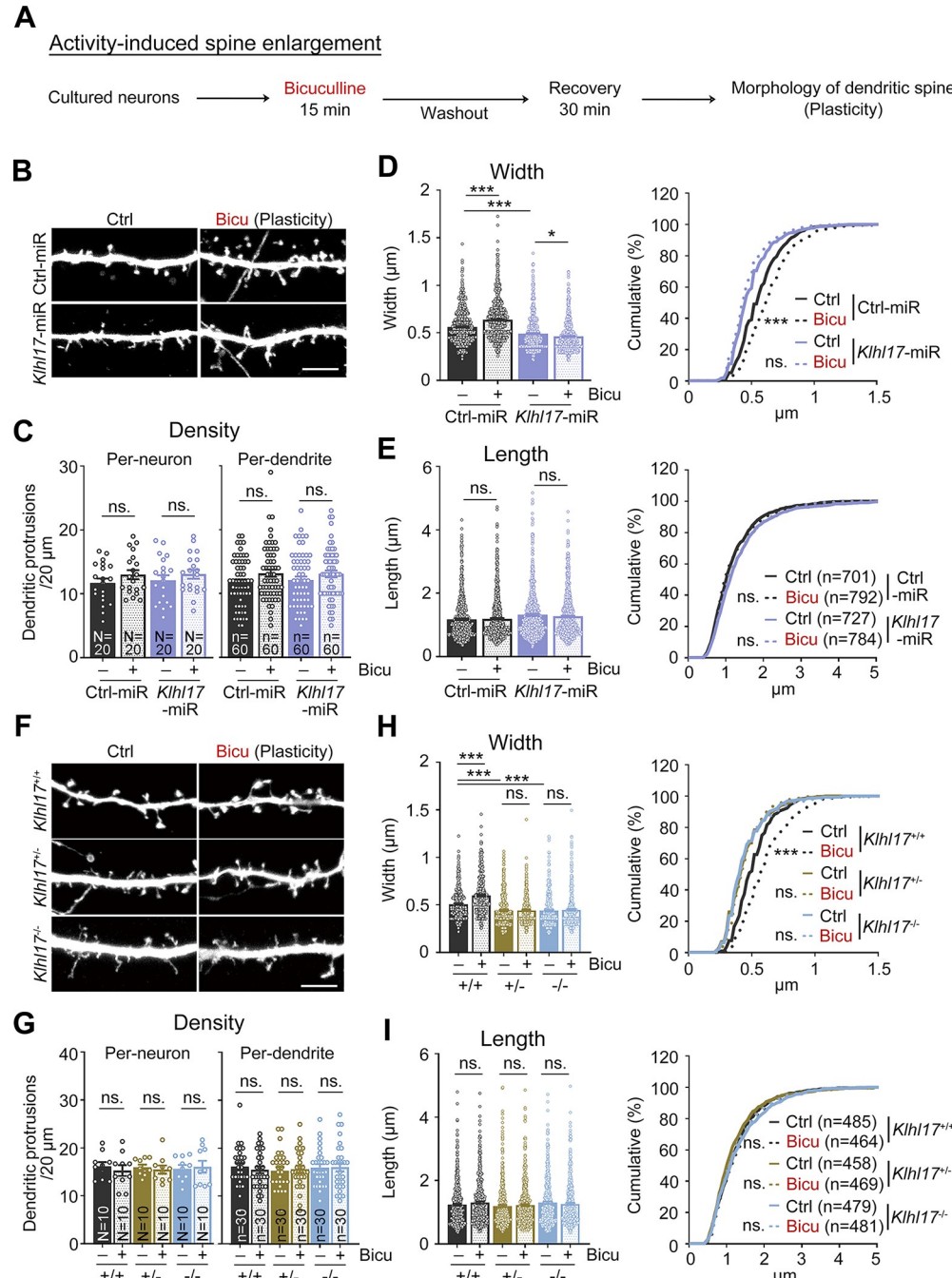

**Fig 2. KLHL17 is required for activity-dependent dendritic spine enlargement.** (**A**) Flowchart of activity-induced spine enlargement. Cultured neurons were transfected with the indicated plasmids at 12 DIV and harvested for immunostaining at 18 DIV. Bicuculline (40 μM) was added into the culture for 15 min and then cultures were subjected to washout for a further 30 min recovery to induce activity-dependent spine enlargement. The morphology of dendritic protrusions was outlined by GFP signals. (**B–E**) *Klhl17* knockdown impairs the synaptic enlargement induced by bicuculline treatment. (**F–I**) Both *Klhl17*$^{+/-}$ and *Klhl17*$^{-/-}$ neurons fail to respond to bicuculline treatment. (**B, F**) Representative images of dendritic segments. (**C, G**) Quantification of protrusion density. Both per-neuron- and per-dendrite-based analyses were performed. (**D, H**) Quantification of the width of dendritic protrusions. (**E, I**) Quantification of the length of dendritic protrusions. Samples were randomly assigned to treatments. The sample size "*N*" indicates the number of examined neurons and "*n*" represents the number of examined dendritic segments in (**C**) and (**G**). In (**E**) and (**I**), the sample size "*n*" indicates the number of dendritic spines examined. For the same sets of experiments, sample sizes are only shown in 1 panel. The data represent mean ± SEM and cumulative curve (**D and E**: right, **H and I**: right). Individual data points are also shown. * $P < 0.05$; *** $P < 0.001$; ns, not significant. Two-way

ANOVA (**C–E**: left, **G–I**: left); Kolmogorov–Smirnov test for cumulative probability (**D and E**: right; **H and I**: right). Scale bars: (**B**, **F**) 5 μm. The numerical value data and statistical results are available in S1 and S2 Data, respectively. DIV, day in vitro; KLHL17, Kelch-like protein 17.

We also performed a paired analysis of calcium events at the spines and dendrites (**Fig 4F**). The events were categorized into 3 groups, i.e., spine-only, dendrite-only, and paired (both spine and dendrite) responses (**Fig 4F**). The percentages of spine-only and dendrite-only events were very low, with a majority ($>95\%$) of paired events in both $Klhl17^{+/-}$ and wild-type neurons under our experimental conditions (**Fig 4G**). In terms of calcium event frequency, only paired events were reduced in $Klhl17^{+/-}$ neurons (**Fig 4H**). For event amplitude, both spine-only and the spine element of paired events exhibited a bigger response in $Klhl17^{+/-}$ neurons relative to wild-type neurons (**Fig 4I**). We also noticed that the amplitudes of both spine- and dendrite-only events were obviously lower than those of paired events (**Fig 4I**). Thus, the differences in calcium events that we recorded are mainly attributable to the paired events, though spine-only events contributed somewhat to differences in amplitude.

Together, these outcomes indicate that KLHL17 regulates both the frequency and amplitude of calcium dynamics in neurons.

## KLHL17 regulates ER distribution in dendritic spines

We then investigated if KLHL17 regulates the spine apparatus. To do so, we transfected cultured neurons with DsRed-ER, Myc-KLHL17, and GFP-actin at 12 DIV to monitor ER according to DsRed-ER signal and KLHL17 distribution in dendritic spines, as outlined by GFP-actin signal, at 18 DIV. We observed that Myc-KLHL17 signal overlapped with that of DsRed-ER in dendritic spines (**Fig 5A and 5B**).

Based on the width of spine heads, we categorized dendritic spines into 2 groups: dendritic spines with heads $>0.5$ μm, i.e., mushroom-like spines, or those $<0.5$ μm, i.e., thin spines. Consistent with the role for the spine apparatus in spine enlargement [33,34], we observed that approximately 60% of mushroom-like spines were ER-positive. For thin spines, only approximately 20% of them contained ER (**Fig 5C and 5D**). KLHL17 protein was also present in approximately 50% of mushroom-like spines (**Fig 5C and 5D**). Importantly, approximately 35% of mushroom-like spines were $KLHL17^+ER^+$, whereas only approximately 5% of thin spines contained both KLHL17 and ER (**Fig 5C and 5D**). Thus, KLHL17 mainly coexisted with ER in dendritic spines possessing larger spine heads.

We then analyzed if $Klhl17$ deficiency alters the ER distribution in dendritic spines. Compared with wild-type neurons, $Klhl17^{+/-}$ and $Klhl17^{-/-}$ neurons displayed a noticeable reduction in ER-positive dendritic spines (**Fig 5E and 5F**). These results indicate that KLHL17 associates with the spine apparatus and controls its distribution in dendritic spines.

## KLHL17 associates with SYNPO and regulates its dendritic spine distribution

The spine apparatus is organized by synaptopodin (SYNPO), an F-actin-binding protein [35]. Thus, SYNPO is recognized as a marker of the spine apparatus and it is known to control ER distribution in dendritic spines [29]. Deletion of $Synpo$ results in a lack of the spine apparatus and impaired synaptic plasticity [36]. Since KLHL17 is also an actin-binding protein, we speculated that it may work with SYNPO to regulate the ER distribution in dendritic spines. In cultured mouse neurons, SYNPO signal indeed colocalized well with that of DsRed-ER under confocal microscopy (**Fig 6A and 6B**, an enlarged image of the entire cell image is presented

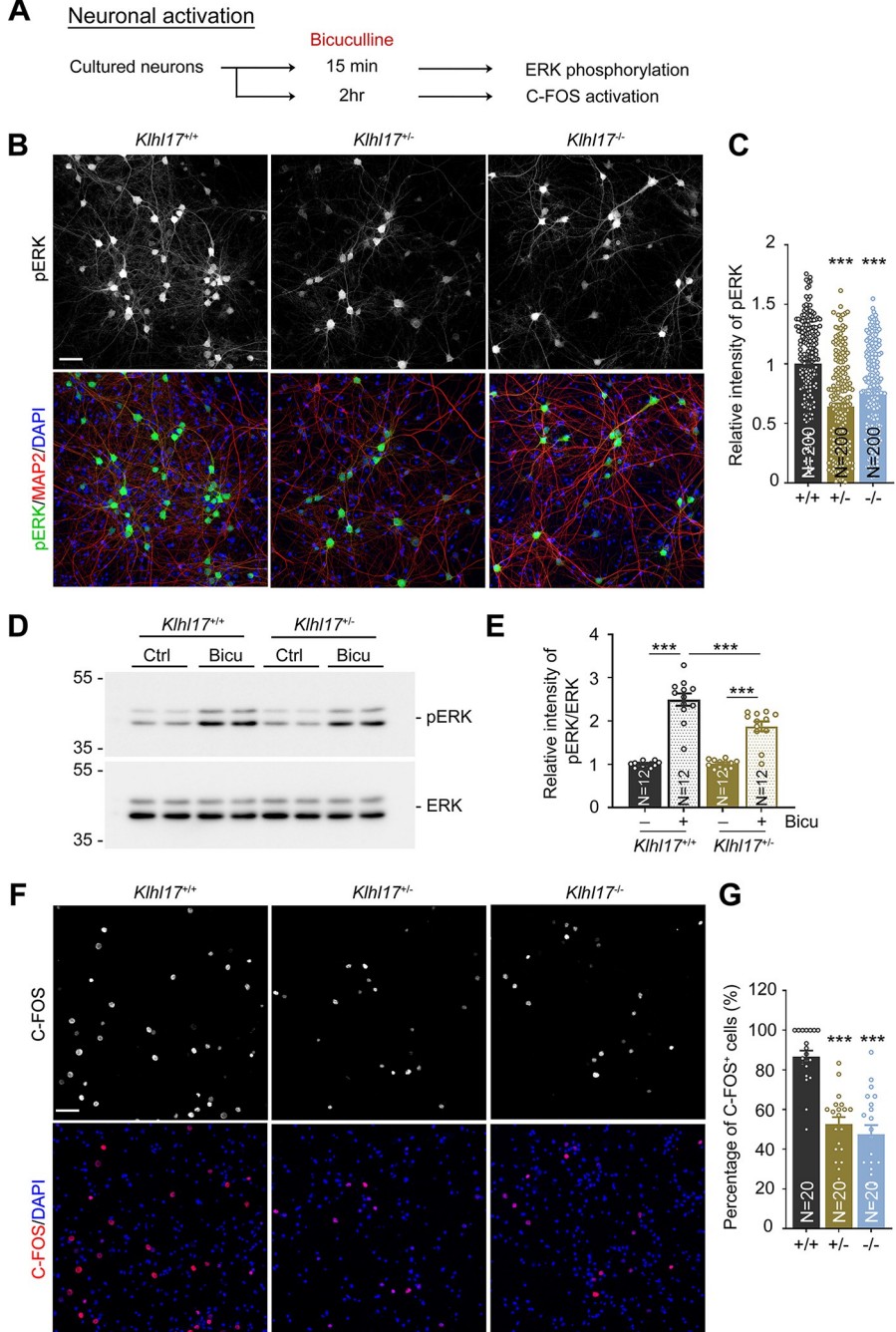

**Fig 3. *Klhl17* deficiency reduces ERK activation and C-FOS expression upon bicuculline treatment.** (**A**) Experimental flowchart. Cultured neurons from different genotypes were treated with bicuculline (40 μM) for 15 min to induce ERK phosphorylation (**B–E**) or 2 h to trigger expression of the immediate early gene C-FOS (**F, G**) at 18 DIV. (**B–E**) Both *Klhl17*[+/−] and *Klhl17*[−/−] neurons exhibit reduced ERK phosphorylation upon bicuculline treatment. (**B**) Representative images of immunostaining. Cultured neurons were stained with the neuronal marker MAP2 and the nuclear marker DAPI. (**C**) Quantification of the relative intensity of phospho-ERK signals based on immunostaining. (**D**) Immunoblots of ERK phosphorylation. (**E**) The quantification of ERK phosphorylation based on immunoblotting. (**F, G**) Numbers of C-FOS-positive cells induced by bicuculline are diminished upon *Klhl17* knockout. (**F**) Representative images. Cultured neurons were stained with C-FOS antibody and DAPI. (**G**) The percentage of C-FOS-positive neurons. Samples were randomly assigned to treatments and collected from 2 independent experiments. The sample size (*N*) indicates the number of analyzed neurons in (**C**), the number of independent preparations in (**E**), and the number of image fields in (**G**). The data represent mean ± SEM. Individual data points are also shown. *** *P* < 0.001; one-way ANOVA. Scale bars: (**B, F**) 50 μm. The numerical value data and

statistical results are available in S1 and S2 Data, respectively. DIV, day in vitro; ERK, extracellular signal-regulated kinase; KLHL17, Kelch-like protein 17.

in S2A Fig). KLHL17 also frequently associated with structures containing both SYNPO and ER (Fig 6A and 6B). We further used LSM980 with AiryScan2 confocal laser scanning and Imaris software to record and analyze fluorescence images, which clearly revealed intermingling of KLHL17, SYNPO, and ER signals at dendritic spines (Fig 6C and 6D, an enlarged image is presented in S2B Fig).

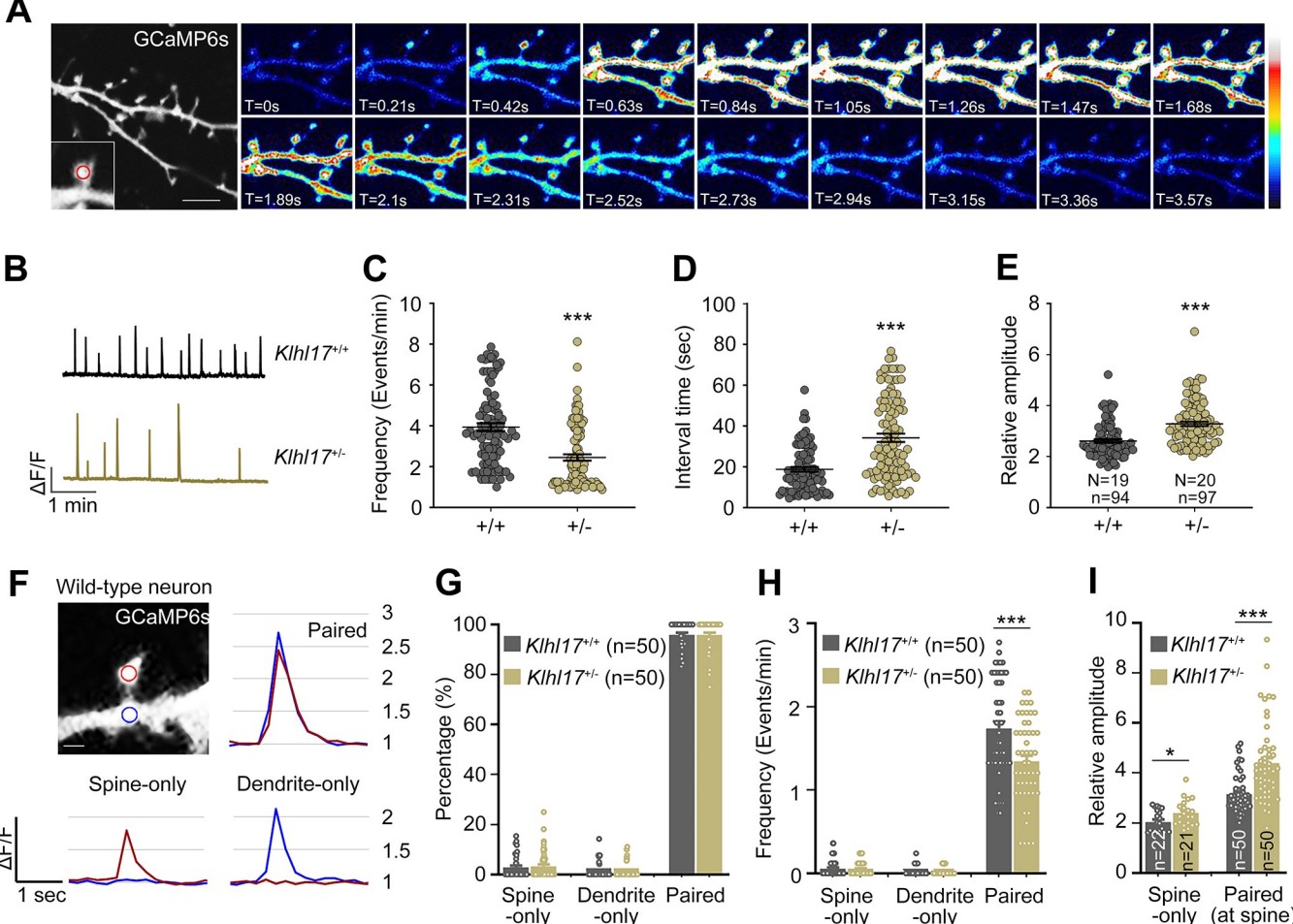

**Fig 4. *Klhl17* deficiency alters the frequency and amplitude of calcium events.** *Klhl17*⁺/⁻ neurons exhibit a reduced frequency and higher amplitude of spontaneous calcium events. Cultured neurons were transfected with GCaMP6s plasmids at 12 DIV and then calcium recording was performed by live-imaging at 18–19 DIV. (**A**) Representative images of recorded frames. Image series showing the change in calcium concentration based on GCaMP6s fluorescence signal intensity. The enlarged segment represents an example of an analyzed dendritic spine and the red circle indicates the ROI. The heat maps show the relative intensities of calcium signals. (**B–E**) Analysis of total calcium events at dendritic spines. (**B**) Representative patterns of calcium events. Scale units: amplitude (ΔF/F) and time (1 min). The calcium dynamics were analyzed based on the parameters of frequency (**C**), interval time (**D**), and calcium amplitude (**E**). Samples were randomly collected from multiple independent experiments. The sample size shown in (**E**) represents the number of examined neurons (*N*) and the number of examined dendritic spines (*n*) for the results shown in (**C–E**). (**F–I**) Paired analysis of calcium events at the spines and dendrites. (**F**) A representative dendritic segment with 2 ROIs; one in the spine (red) and the other in the dendrite (blue). Calcium events were categorized into 3 types: spine-only, dendrite-only, and both spine and dendrite with spikes at the same time (paired). Examples of these 3 types of events from wild-type neurons are shown. The relative amplitude (1, 1.5, 2, 2.5, 3) is also indicated at right. The percentage (**G**), frequency (**H**), and amplitude (**I**) of these 3 types of events are shown. The number (*n*) of examined paired ROIs was 50. In (**I**), the amplitude of all detected spine-only events was analyzed. The sample size is indicated. For paired events, the average amplitude of the paired ROIs at spines is shown. The data represent mean ± SEM. Individual data points are also shown. * $P < 0.05$, *** $P < 0.001$; unpaired two-tailed *t* test. Scale bars: (**A**) 5 μm, (**F**) 1 μm. The numerical value data and statistical results are available in S1 and S2 Data, respectively. DIV, day in vitro; KLHL17, Kelch-like protein 17; ROI, region of interest.

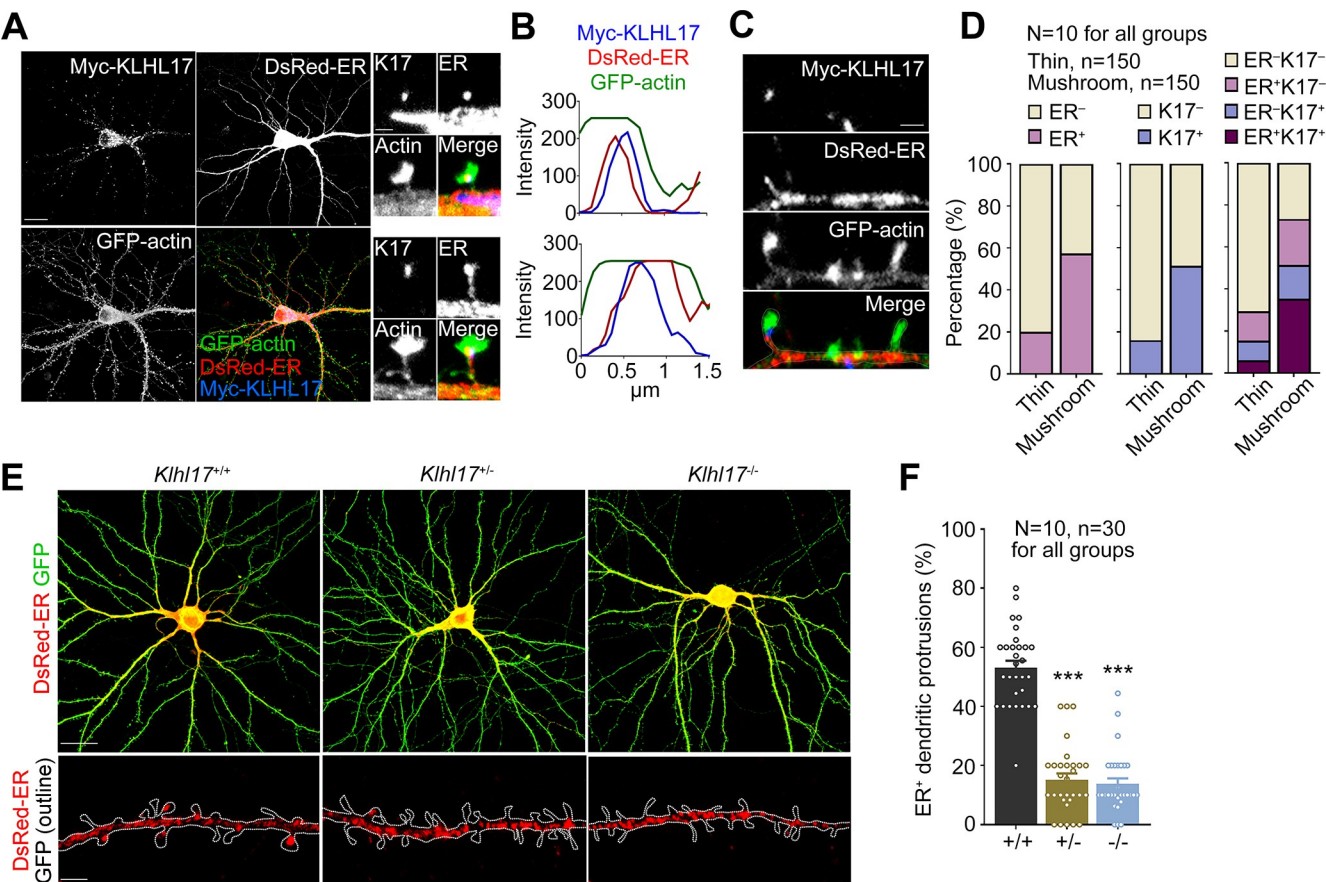

**Fig 5.** *Klhl17* **deficiency impairs the dendritic spine distribution of the ER.** (**A**, **B**) Overlapping distributions of KLHL17 and ER in dendritic spines. Cultured neurons were co-transfected with a Myc-KLHL17 and DsRed-ER construct at 12 DIV and immunostained at 18 DIV. (**A**) Confocal images of a neuron and enlarged confocal images of 2 dendritic spines. (**B**) Line scan of the enlarged dendritic spines shown in (**A**). KLHL17 and ER signals overlap along the dendritic spines. (**C**, **D**) KLHL17 and ER tend to be present in dendritic spines with larger heads (>0.5 μm), i.e., mushroom-like spines indicated as "mushroom." "Thin" represents dendritic spines with spine heads <0.5 μm. (**C**) Representative images showing the distributions of KLHL17 and DsRed-ER in thin and mushroom-like spines. (**D**) Quantification of the percentage of KLHL17$^+$ and ER$^+$ protrusions in thin and mushroom-like spines. (**E**, **F**) *Klhl17*-deficient neurons have fewer ER-positive dendritic spines. Cultured neurons from different genotypes were transfected with the indicated plasmids at 12 DIV and harvested for immunostaining at 18 DIV. (**E**) Representative images of neurons and enlarged dendrite segments. (**F**) Quantification of the percentage of ER$^+$ dendritic spines. Images were randomly collected from 2 independent experiments. The sample size "*N*" indicates the number of examined neurons and "*n*" represents the number of examined dendritic spines (**D**) or examined dendritic segments (**F**). The data represent mean ± SEM. Individual data points are also shown. *** *P* < 0.001; one-way ANOVA. Scale bars: (**A**) 20 μm; (**C**) 1 μm; (**E**) whole cells: 20 μm; enlarged segment: 2 μm. The numerical value data and statistical results are available in S1 and S2 Data, respectively. DIV, day in vitro; ER, endoplasmic reticulum; KLHL17, Kelch-like protein 17.

In neuroblastoma Neuro-2A cells, immunoprecipitation of Myc-KLHL17 co-precipitated HA-SYNPO (**Fig 6E**). Although co-immunoprecipitation efficiency was not high, it was specific because Myc antibody did not precipitate HA-SYNPO in the absence of Myc-KLHL17 (**Fig 6E**).

We further investigated if KLHL17 influences dendritic spine targeting of SYNPO. We found that *Klhl17*$^{+/−}$ and *Klhl17*$^{−/−}$ neurons both had noticeably fewer dendritic spines containing SYNPO (**Fig 6F and 6G**). Given that SYNPO is a marker of the spine apparatus, this outcome is consistent with the effect of *Klhl17* deficiency on ER distribution (**Fig 5E and 5F**). Moreover, considering the size of spine heads, *Klhl17* deficiency primarily reduced the percentage of SYNPO$^+$ mushroom-like spines and increased that of SYNPO$^−$ thin spines (**Fig 6H**), supporting the notion that KLHL17 regulates the dendritic spine distribution of SYNPO and controls dendritic spine enlargement.

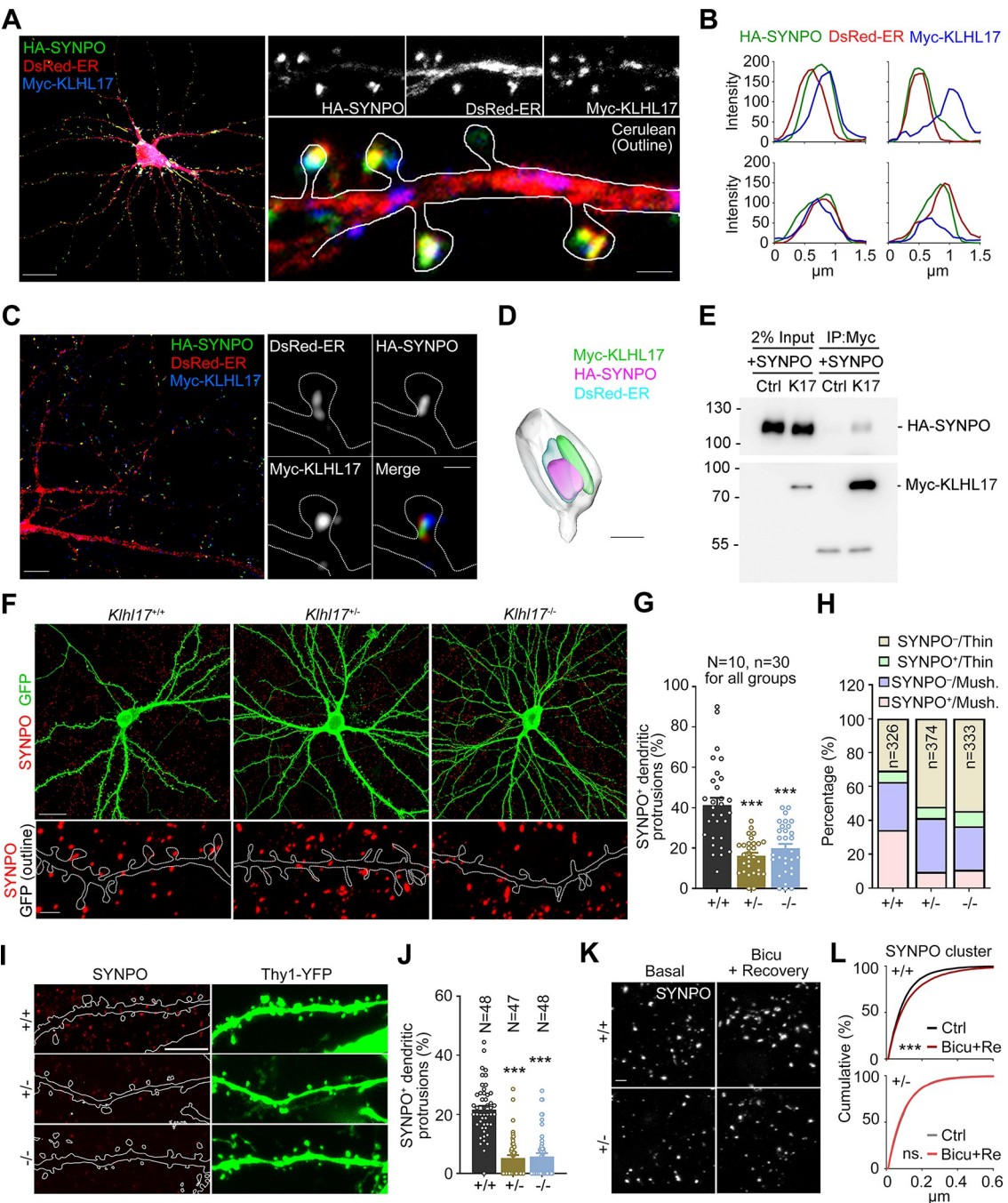

**Fig 6. *Klhl17* deficiency impairs synaptic expression and the distribution of synaptopodin.** (**A–D**) Colocalization or adjacent distributions of SYNPO and KLHL17 in cultured neurons. Cultured neurons were co-transfected with Myc-KLHL17, DsRed-ER, HA-SYNPO, and Cerulean constructs at 12 DIV and immunostained at 18 DIV. (**A**) Confocal images of the whole cell and enlarged dendrite segments. (**B**) Line scan results for Myc-KLHL17, DsRed-ER, and HA-SYNPO in the dendritic spines. (**C**) Super-resolution images of the whole cell and enlarged dendritic spines. (**D**) 3D reconstruction image using Imaris. (**E**) Coimmunoprecipitation of KLHL17 and SYNPO in Neuro-2A neuroblastoma cells. Neuro-2A cells were cotransfected with Myc-KLHL17 and HA-SYNPO constructs and subjected to immunoprecipitation and immunoblotting using the indicated antibodies. (**F–H**) *Klhl17*-deficient neurons in culture have fewer SYNPO-positive dendritic spines. (**F**) Representative images of the whole cells and enlarged dendrite segments. (**G**) Quantification of (**F**), i.e., the percentage of SYNPO⁺ dendritic spines. (**H**) Quantification of the percentage of SYNPO⁺ protrusions in thin and mushroom-like (mush.) spines of *Klhl17*⁺/⁺, *Klhl17*⁺/⁻, and *Klhl17*⁻/⁻ neurons. (**I, J**) The percentage of SYNPO-positive dendritic spines in vivo is also reduced by *Klhl17* deficiency. CA1 neurons of mice expressing YFP driven by the Thy1 promoter were analyzed. (**I**) Representative images. (**J**) The percentage of SYNPO⁺ protrusions. (**K, L**) Activity-dependent SYNPO clustering. Cultured neurons were treated with bicuculline (40 μM) for 15 min and then underwent washout for recovery for

a further 30 min, which altered activity-dependent SYNPO clustering. (**K**) Representative images of endogenous SYNPO immunoreactivity. (**L**) Quantification of the size of SYNPO puncta upon synaptic stimulation. Samples were randomly assigned to treatments and collected from 2 independent experiments. In (**G**) and (**J**), the sample size "*N*" indicates the number of examined neurons. In (**G**), "*n*" represents the number of examined dendritic segments. In (**H**), "*n*" represents the number of examined dendritic spines. In (**L**), 10 image fields for each group were randomly collected from 2 independent experiments. All SYNPO-positive puncta (*Klhl17*$^{+/+}$/Ctrl: 6523; *Klhl17*$^{+/+}$/Bicu+Re: 7680; *Klhl17*$^{+/−}$/Ctrl: 5354; *Klhl17*$^{+/−}$/Bicu+Re: 6672) in the images were analyzed. The data represent mean ± SEM. Individual data points are also shown. *** $P < 0.001$; ns, not significant. One-way ANOVA (**G**, **J**); Kolmogorov–Smirnov test for cumulative probability (**L**). Scale bars: (**A**) whole cells: 20 μm; enlarged segment: 1 μm; (**C**) whole cells: 5 μm; enlarged segment: 0.5 μm; (**D**) 0.5 μm; (**F**) whole cells: 20 μm; enlarged segment: 2 μm; (**I**) 5 μm; (**K**) 2 μm. The numerical value data and statistical results are available in S1 and S2 Data, respectively. DIV, day in vitro; KLHL17, Kelch-like protein 17.

Apart from our aforementioned experiments on cultured neurons, we also examined CA1 neurons in mouse brains. Using Thy1-YFP to outline neuronal morphology, including dendritic spines, we detected fewer SYNPO$^+$ spines in both *Klhl17*$^{+/−}$and *Klhl17*$^{−/−}$mouse brains compared to their wild-type littermates (**Fig 6I and 6J**).

Related to its function in organizing the spine apparatus, SYNPO is also recognized as a plasticity protein that senses divergences from basal activity levels to maintain the homeostasis of synaptic activity [37,38], implying that activity-dependent SYNPO remodeling (i.e., clustering) likely acts as a regulatory mechanism to adjust the strength of excitatory synapses and modulate network activity. Accordingly, we investigated if SYNPO is also regulated by KLHL17 in an activity-dependent manner. To do so, we compared SYNPO clustering in wild-type and *Klhl17*$^{+/−}$neurons upon bicuculline treatment for 15 min followed by 30 min recovery. We found that after stimulation, wild-type neurons, but not *Klhl17*$^{+/−}$neurons, displayed larger SYNPO puncta (**Fig 6K and 6L**), indicating that *Klhl17* deficiency also impairs SYNPO clustering upon neural activation.

Taken together, these findings support that KLHL17 controls SYNPO clustering at dendritic spines to facilitate synaptic potentiation.

## SYNPO mediates the effect of KLHL17 on calcium dynamics, spine enlargement, and neuronal activation

Based on our findings, we hypothesized that KLHL17 regulates ER distribution via SYNPO, thereby controlling calcium release from the ER to dendritic spines and consequently promoting dendritic spine enlargement and neuronal activation. To investigate that speculation, we first confirmed the involvement of the spine apparatus in the calcium events controlled by KLHL17. To do so, we treated neurons with ryanodine to inhibit calcium efflux from ER via the ryanodine receptor localized on ER. Ryanodine treatment of wild-type neurons reduced the frequency and increased the interval of calcium events at dendritic spines (**Fig 7A–7C**). The amplitude of calcium events was not obviously altered, perhaps because other types of calcium channels were involved (**Fig 7D**). *Klhl17*$^{−/−}$neurons exhibited a reduced frequency but an increased amplitude of calcium events (**Fig 7A–7D**). Importantly, ryanodine treatment did not further alter the frequency or amplitude of the calcium events of *Klhl17*$^{−/−}$neurons (**Fig 7A–7D**). Thus, this insensitivity of *Klhl17*$^{−/−}$neurons to ryanodine treatment is consistent with our observation that *Klhl17* deficiency impairs the synaptic distribution of ER.

We then examined if SYNPO overexpression ameliorates *Klhl17* deficiency. In terms of calcium dynamics, we found that SYNPO overexpression improved all of the deficits of calcium dynamics prompted by *Klhl17* deficiency, including the frequency, interval time, and amplitude of calcium events in cultured neurons (**Fig 7E–7H**).

For the features of dendritic spines, we observed that SYNPO overexpression did not alter the density or length of dendritic spines, but it did specifically enhance the width of *Klhl17*$^{+/}$

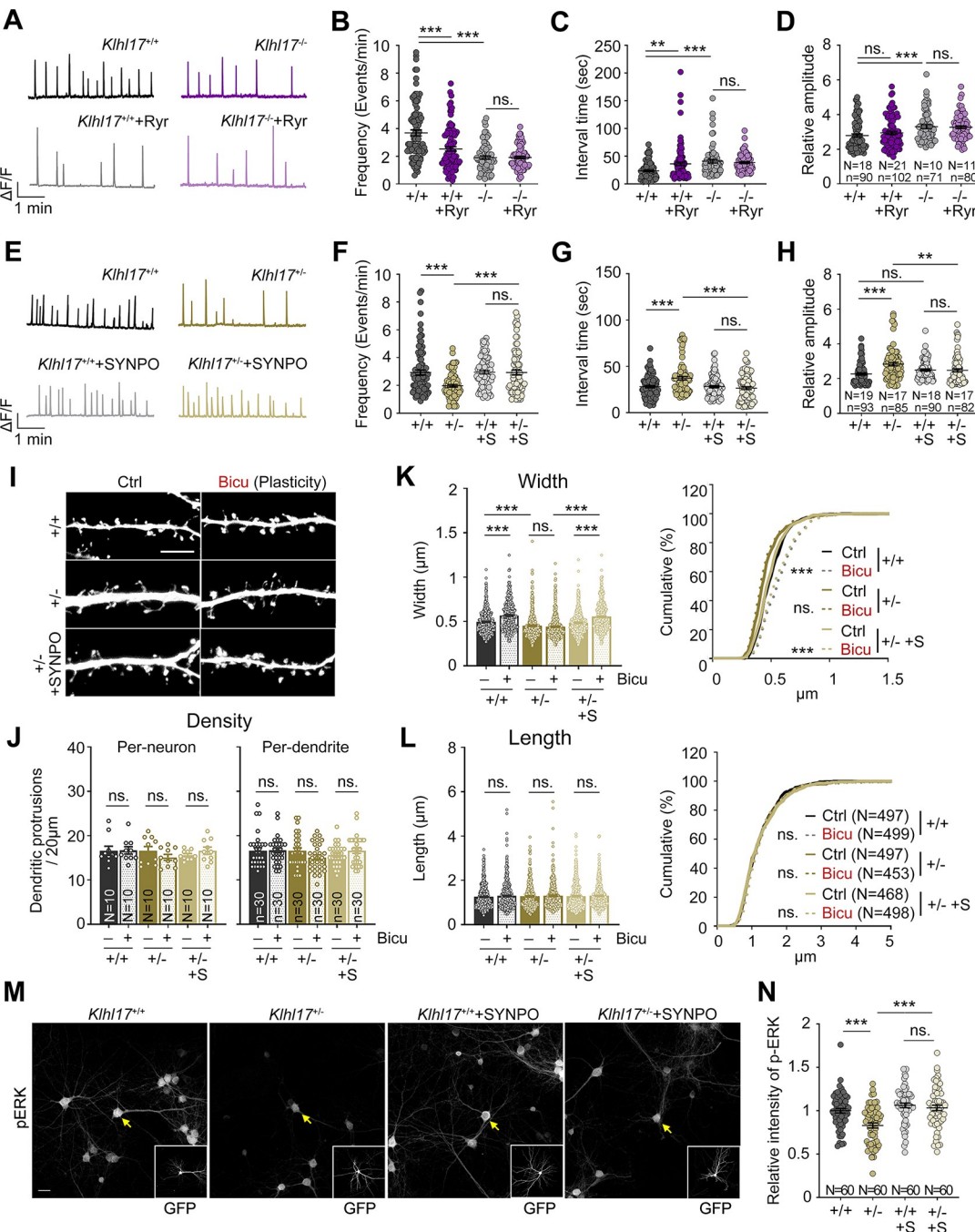

**Fig 7. SYNPO overexpression rescues the deficits of *Klhl17*-deficient neurons.** (**A–D**) Ryanodine treatment alters calcium dynamics in wild-type neurons but not *Klhl17^(−/−)* neurons. (**A**) Representative patterns of calcium events. Scale units: amplitude (ΔF/F) and time (1 min). Quantification of the frequency (**B**), interval time (**C**), and amplitude (**D**) of calcium events. (**E–H**) SYNPO overexpression restores calcium dynamics in *Klhl17*-deficient neurons. Cultured neurons were co-transfected with GCaMP6s and SYNPO (or "S") or vector control, as indicated, at 12 DIV, before performing calcium recording by means of live-imaging at 18–19 DIV. (**E**) Representative patterns of the calcium events. Scale units: amplitude (ΔF/F) and time (1 min). Quantification of frequency (**F**), interval time (**G**) and amplitude (**H**). (**I–L**) SYNPO overexpression also rescues the spine enlargement of *Klhl17*-deficient neurons upon bicuculline stimulation. Cultured neurons were transfected with GFP and SYNPO or vector control at 12 DIV, as indicated, and performed activity-induced spine enlargement at 18 DIV. (**I**) Representative images of dendritic segments. (**J**) Quantification of density, (**K**) width and (**L**) length of dendritic protrusions. In (**J**), the density of dendritic spines was analyzed on both a per-neuron and per-dendrite basis. (**M, N**) SYNPO overexpression increases ERK phosphorylation of *Klhl17*-deficient neurons upon bicuculline treatment. Cultured neurons were co-transfected with GFP and SYNPO or vector control at 12 DIV and treated with bicuculline at 18 DIV. (**M**) Representative images. GFP

images are shown in insets, with transfected cells highlighted by yellow arrows. (**N**) Quantification of the relative intensity of phospho-ERK signals. All data were collected from at least 2 independent experiments. The sample sizes of examined neurons (*N*), dendritic segments (*n*), and dendritic spines (*n*) are indicated. For the same set of experiments, the sample size is labeled only in 1 panel. The data represent mean ± SEM and cumulative curves (**K, L**: right). Individual data points are also shown. ** $P < 0.01$; *** $P < 0.001$; ns, not significant. Two-way ANOVA (**B–D, F–H, J–L**: left, **N**); Kolmogorov–Smirnov test for cumulative probability (**K, L**: right). Scale bars: (**I**) 5 μm; (**M**) 20 μm. The numerical value data and statistical results are available in S1 and S2 Data, respectively. DIV, day in vitro; ERK, extracellular signal-regulated kinase; KLHL17, Kelch-like protein 17.

⁻dendritic spines to values comparable to wild-type neurons (**Fig 7I–7L**). When cultured neurons were activated by means of bicuculline treatment, the spine width of SYNPO-overexpressing *Klhl17*⁺/⁻ neurons was also fully rescued to the levels of wild-type neurons subjected to the same treatment (**Fig 7I and 7K**). These results reveal that SYNPO is involved in KLHL17-controlled dendritic spine enlargement. Moreover, the effect is specific because neither the density nor length of dendritic spines was affected upon SYNPO overexpression (**Fig 7I–7L**).

Finally, SYNPO overexpression also improved neuronal activation of *Klhl17*⁺/⁻ neurons because it enhanced ERK phosphorylation in *Klhl17*⁺/⁻ neurons to levels comparable to wild-type neurons (**Fig 7M and 7N**).

Thus, SYNPO overexpression ameliorates the perturbed calcium dynamics, dendritic spine enlargement, and neuronal activation of *Klhl17*-deficient neurons, supporting the involvement of SYNPO in KLHL17-dependent regulation.

## KLHL17 and SYNPO association is critical for ER distribution and calcium dynamics

Next, we investigated how KLHL17 and SYNPO associate with each other. KLHL17 contains an N-terminal BTB domain and C-terminal Kelch actin-binding domains [11]. SYNPO contains two 14-3-3 binding motifs and a region involved in actinin interaction [22,39]. We constructed truncated proteins containing either the N- or C-terminal regions of KLHL17 and SYNPO (**Fig 8A**) and investigated which regions are involved in the KLHL17 and SYNPO interaction. Although the co-immunoprecipitation efficiency between KLHL17 and SYNPO is low, similar to the interaction between full-length proteins, we still observed specific interaction between the N-terminal fragment of KLHL17 (i.e., K17-N) and full-length SYNPO (**Fig 8B**) and between the C-terminal region of SYNPO (i.e., SYNPO-C) and full-length KLHL17 (**Fig 8C**), whereas N-terminal SYNPO and C-terminal KLHL17 were not associated with full-length KLHL17 or SYNPO, respectively.

To further confirm that the association between KLHL17 and SYNPO is critical for KLHL17 functionality, we overexpressed SYNPO-C in wild-type cultured neurons, which we anticipated would disrupt the interaction between endogenous KLHL17 and SYNPO. Vector control and SYNPO-N were included in the experiment to compare the effect of SYNPO-C. SYNPO-C formed clusters in cultured neurons (**Fig 8D**), as also observed for full-length SYNPO. In contrast, SYNPO-N was more evenly distributed across neurons (**Fig 8D**). Moreover, we found that the dendritic spine distribution of ER was reduced by SYNPO-C overexpression, but that was not the case for SYNPO-N (**Fig 8D and 8E**). Furthermore, the percentages of both ER⁺ mushroom-like and ER⁺ thin spines were reduced upon SYNPO-C overexpression (**Fig 8F**). These results indicate that the C-terminal region of SYNPO is involved in regulating the distribution of the spine apparatus.

We then expressed SYNPO-C in *Klhl17*⁻/⁻ neurons. We found that SYNPO-C did not further reduce the distribution of ER in *Klhl17*⁻/⁻ neurons (**Fig 8G and 8H**). The percentages of ER⁺ spines were comparable among the groups of *Klhl17*⁻/⁻ neurons expressing SYNPO-C,

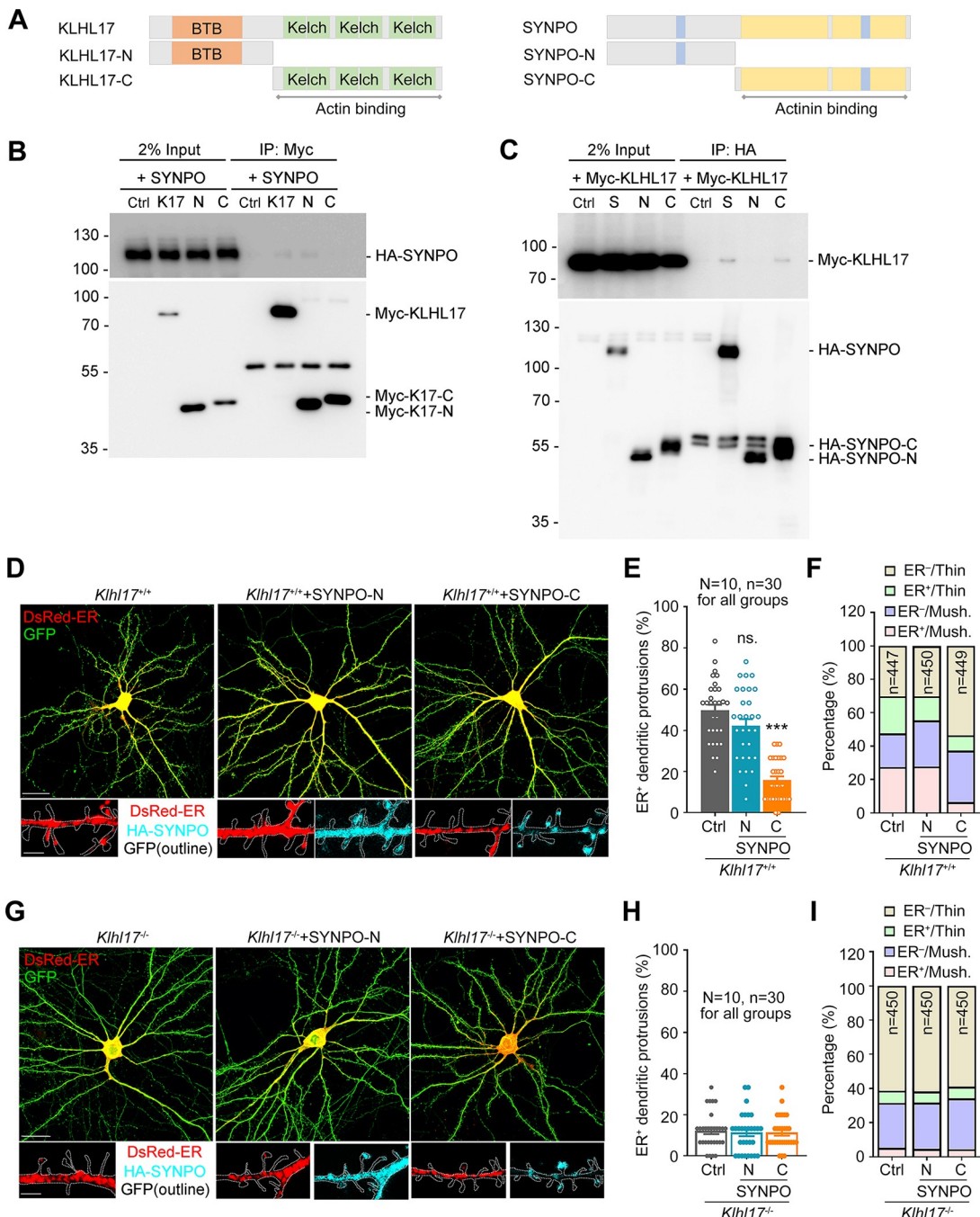

**Fig 8. KLHL17 and SYNPO association controls the dendritic spine distribution of the ER.** (**A**) Schematic domain structures of KLHL17 and SYNPO. Left: Full-length KLHL17 and truncated KLHL17-N (amino acid residues 1–300) and KLHL17-C (amino acid residues 301–640) mutants. KLHL17-N contains the BTB domain, whereas KLHL17-C comprises 6 Kelch domains. Right: Full-length SYNPO and truncated SYNPO-N (amino acid residues 1–299) and SYNPO-C (amino acid residues 300–690) mutants. SYNPO-C includes 2 regions required for the actinin-binding domain (yellow blocks) and one 14-3-3 binding motif (blue block). The other 14-3-3 binding motif is located in the N-terminal region of SYNPO. (**B, C**) C-terminal SYNPO (SYNPO-C) co-immunoprecipitates with N-terminal KLHL17 (KLHL17-N). Neuro-2A cells were cotransfected with either full-length Myc-KLHL17 or HA-SYNPO with the N- or C-terminal constructs, as indicated, and subjected to immunoprecipitation and immunoblotting using the indicated antibodies. (**D**–**F**) Overexpression of SYNPO-C reduces the number of ER-positive dendritic spines of wild-type neurons. Cultured neurons were transfected with the indicated plasmids at 12 DIV and harvested for immunostaining at 18 DIV. (**D**) Representative images of the whole cells and enlarged dendrite segments. SYNPO-C exhibits a punctate signal like that of full-length SYNPO, whereas SYNPO-N shows a diffuse pattern. (**E**) Quantification of the percentage of ER$^+$ dendritic spines. (**F**) Quantification of the percentage of ER$^+$ protrusions in thin and mushroom-like (Mush.) spines. (**G**–

I) The same experiments as (**D**–**F**), except that *Klhl17*<sup>−/−</sup> neurons were used. In (**E**) and (**H**), the sample sizes of examined neurons (*N*) and dendritic segments (*n*) are indicated. In (**F**) and (**I**), the numbers of dendritic spines (*n*) examined are indicated. All samples were collected from 2 independent experiments. The data represent mean ± SEM. Individual data points are also shown. *** $P < 0.001$; ns, not significant; one-way ANOVA. Scale bars: (**D**), (**G**) whole cells: 20 μm; enlarged segment: 2 μm. The numerical value data and statistical results are available in S1 and S2 Data, respectively. BTB, Bric-a-brac/Tramtrack/Broad; DIV, day in vitro; ER, endoplasmic reticulum; KLHL17, Kelch-like protein 17.

SYNPO-N, and control vector (**Fig 8G–8I**). These results further strengthen the evidence for involvement of SYNPO-C in the function of KLHL17 in controlling synaptic ER distribution.

Similar to the effect of *Klhl17* deficiency on calcium dynamics, expression of SYNPO-C, but not SYNPO-N, reduced the frequency of and increased the interval between calcium events (**Fig 9A–9C**), consistent with the notion that KLHL17 associates with the C-terminal region of SYNPO to regulate the frequency of calcium influx. However, in contrast to *Klhl17* deficiency that increased the amplitude of calcium influx, overexpression of both SYNPO-C and SYNPO-N reduced the amplitude of calcium events (**Fig 9D**), supporting that another mechanism is involved in how SYNPO controls the amplitude of calcium events in neurons.

We further investigated the effect of SYNPO-C on dendritic spine morphology. Similar to *Klhl17* deficiency, SYNPO-C overexpression did not alter the density or length of dendritic spines, but specifically reduced the width of the dendritic spines of wild-type cultured neurons (**Fig 9E–9H**). SYNPO-C overexpression also impaired activity-dependent dendritic spine enlargement (**Fig 9E–9G**). In terms of neuronal activation, overexpression of SYNPO-C, but not SYNPO-N, also reduced ERK phosphorylation in wild-type neurons upon bicuculline treatment (**Fig 9I and 9J**).

Thus, the effect of SYNPO-C overexpression is generally similar to that arising from *Klhl17* deficiency in terms of regulating the synaptic ER distribution, the frequency of calcium events, dendritic spine enlargement, and neuronal activation. These results support the idea that KLHL17 associates with the C-terminal region of SYNPO to control synaptic plasticity.

## Expansion microscopy further reveals the role of KLHL17 in synaptic ER organization

Finally, we conducted super-resolution imaging using expansion microscopy to further characterize associations among KLHL17, SYNPO, and ER, as well as the organization of the spine apparatus. To do so, we modified a published protocol [40] by replacing 2 reagents [41]. First, we replaced an oligonucleotide-linked secondary antibody with a conventional fluorophore-conjugated secondary antibody. Second, we substituted proteinase K with trypsin to digest post-stained neurons embedded in a hydrogel [41]. Proteinase K is a potent and broad-spectrum protease that cleaves peptide bonds at the carboxylic terminal of aromatic, aliphatic, or hydrophobic amino acids. In contrast, trypsin specifically cleaves at the C-terminal side of Arg and Lys. Therefore, trypsin digestion is expected to retain more fluorophore-conjugated peptides in the hydrogel. We measured the average diameter of somata before and after expansion (**Fig 10A,** pre-Ex versus post-Ex), with average diameter increasing from 15.38 μm to 53.91 μm, i.e., 3.5-fold (**Fig 10A**). Complete and continuous neuronal morphology was clearly observable after expansion (**Fig 10A and 10B**). Moreover, fine-scale structures of dendritic spines, filopodia, and axons were well preserved (**Fig 10B**). Thus, cultured neurons were isotropically expanded under our modified protocol. Since the resolution of our LSM980 with Airyscan 2 confocal laser scanner is 90 to 120 nm (depending on whether Airyscan Joint Deconvolution is applied), the resolution of our imaging system may be up to 26 to 35 nm after expansion.

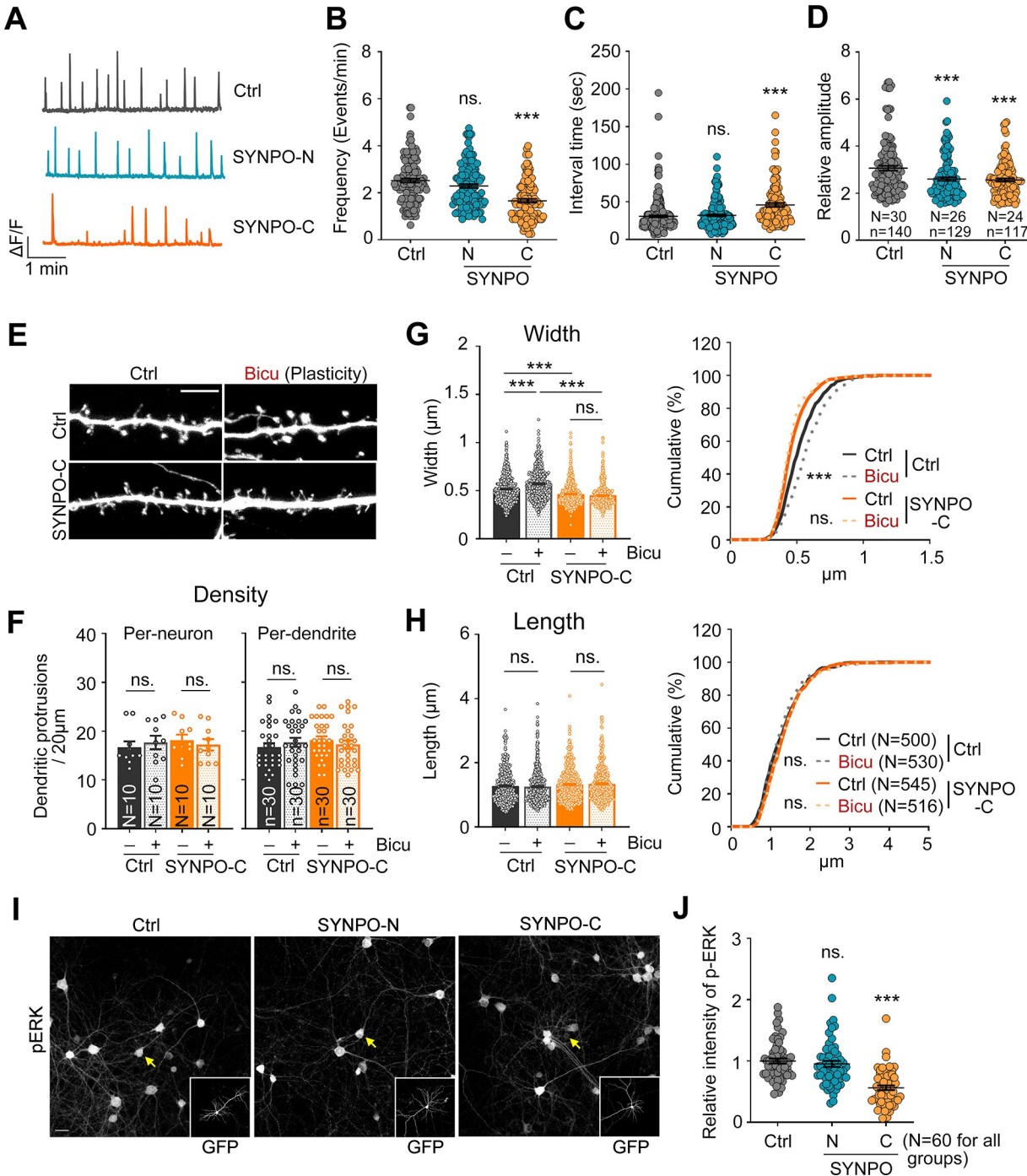

**Fig 9. The KLHL17-SYNPO association is required for activity-dependent dendritic spine enlargement and neuronal activation.** (**A–D**) Overexpression of SYNPO-C impairs calcium events. Cultured neurons were transfected with GCaMP6s and the indicated plasmids at 12 DIV before performing calcium recording based on GCaMP6s signals using live-imaging at 18–19 DIV. (**A**) Representative patterns of calcium events. Scale units: amplitude (ΔF/F) and time (1 min). Quantification of frequency (**B**), interval time (**C**), and amplitude (**D**). (**E–H**) Overexpression of SYNPO-C impairs the spine enlargement induced by bicuculline. Cultured neurons were transfected with the indicated plasmids at 12 DIV and activity-induced spine enlargement was assessed at 18 DIV. (**E**) Representative images of dendritic segments. (**F**) Quantification of density, (**G**) width, and (**H**) length of dendritic protrusions. In (**F**), the density of dendritic spines was analyzed on both a per-neuron and per-dendrite basis. (**I, J**) Overexpression of SYNPO-C reduces bicuculline-induced ERK phosphorylation. Cultured neurons were transfected with the indicated plasmids at 12 DIV and treated with bicuculline at 18 DIV. (**I**) Representative images. GFP images are shown in insets, with yellow arrows indicating transfected neurons. (**J**) Quantification of the relative intensity of phospho-ERK signals. Images were collected from 2 independent experiments. All data were collected from at least 2 independent experiments. The sample sizes of examined

neurons (*N*), dendritic segments (*n*), and dendritic spines (*n*) are indicated. For the same set of experiments, the sample size is labeled only in 1 panel. The data represent mean ± SEM and cumulative curves (**G**, **H**: right). Individual data points are also shown. *** $P < 0.001$; ns, not significant. One-way ANOVA (**B**–**D**, **F**–**H**: left; **J**); Kolmogorov–Smirnov test for cumulative probability (**G**, **H**: right). Scale bars: (**E**) 5 μm; (**I**) 20 μm. The numerical value data and statistical results are available in S1 and S2 Data, respectively. DIV, day in vitro; ERK, extracellular signal-regulated kinase; KLHL17, Kelch-like protein 17.

Using this system, we monitored the distribution of KLHL17, SYNPO, and ER in dendritic spines. The super-resolution images generated by expansion microscopy were then processed using Imaris. The resulting images showed that SYNPO and ER signals were always intermingled with each other and that KLHL17 was adjacent to the SYNPO/ER complex or at the front of the complex toward the dendritic spine tip (**Fig 10C**). Only a very small proportion of KLHL17 signals were intermingled with ER and SYNPO (**Figs 10C and S3**). Our 3D images also revealed complex structures containing KLHL17, SYNPO, and ER (**Fig 10C**, right).

Since our imaging resolution increased approximately 3.5-fold, we also noticed some alterations to the spine apparatus of *Klhl17*$^{+/-}$ neurons. Some ER tended to cluster together, but some was separated from each other, even within dendritic spines (**Fig 10D**). Accordingly, we categorized the distribution patterns of the spine apparatus into 4 groups, i.e., cluster, sparse, neck, or none. "Cluster" reflected aggregation of ER tubules at dendritic spines. "Sparse" represented ER tubules that were separate from each other within dendritic spines. "Neck" represented that ER tubules were solely distributed along the neck of dendritic spines. Finally, "None" encompassed the scenario when no ER signal was detected within dendritic spines or the neck (**Fig 10D**). Similar to our above-described conclusions, *Klhl17* deficiency and SYNPO-C expression reduced the percentage of ER-positive dendritic spines after expansion (**Fig 10E**). Given that the filopodia and thin spines usually do not contain ER, we only analyzed spines with heads >2 μm after expansion (i.e., >~0.57 μm before expansion) in the analysis of ER morphology. The percentage of spines containing the "cluster" type of ER was clearly reduced in *Klhl17*$^{+/-}$ neurons and SYNPO-C-expressing neurons, with the percentages of dendritic spines lacking ER also being increased in these 2 neuronal types (**Fig 10F**). The "sparse" type of ER was slightly increased among *Klhl17*$^{+/-}$ neurons and SYNPO-C-expressing neurons (**Fig 10F**). Thus, our super-resolution imaging analyses suggest that KLHL17 and SYNPO act together to control the clustering and distribution of ER within dendritic spines.

## Discussion

Herein, we have demonstrated roles for KLHL17 and SYNPO, 2 autism-linked proteins, in synaptic plasticity. We have shown that protein levels of KLHL17 are increased by neuronal activation through NMDAR signaling and protein synthesis. KLHL17 controls activity-dependent dendritic spine enlargement and is critical for neuronal activation. We have further demonstrated that KLHL17 works together with SYNPO to control the synaptic distribution of the spine apparatus, thereby altering calcium dynamics at dendritic spines and consequently regulating dendritic spine enlargement and neuronal activation. Thus, our findings reveal for the first time that KLHL17 is required for neuronal activation by controlling synaptic ER distribution and calcium dynamics. Given that KLHL17 protein levels are up-regulated by neuronal activation, a positive feedback regulation apparently contributes to maintaining the expression and function of KLHL17 in neurons.

Regulation of KLHL17 expression is complex. Although *Klhl17* RNA levels tended to be reduced during development and upon neuronal activation in our cultured neurons (**Fig 1A and 1E**), KLHL17 protein levels actually increased. It remains unclear how *Klhl17* RNA levels are controlled. Perhaps transcriptional regulation or RNA stability is involved, especially given

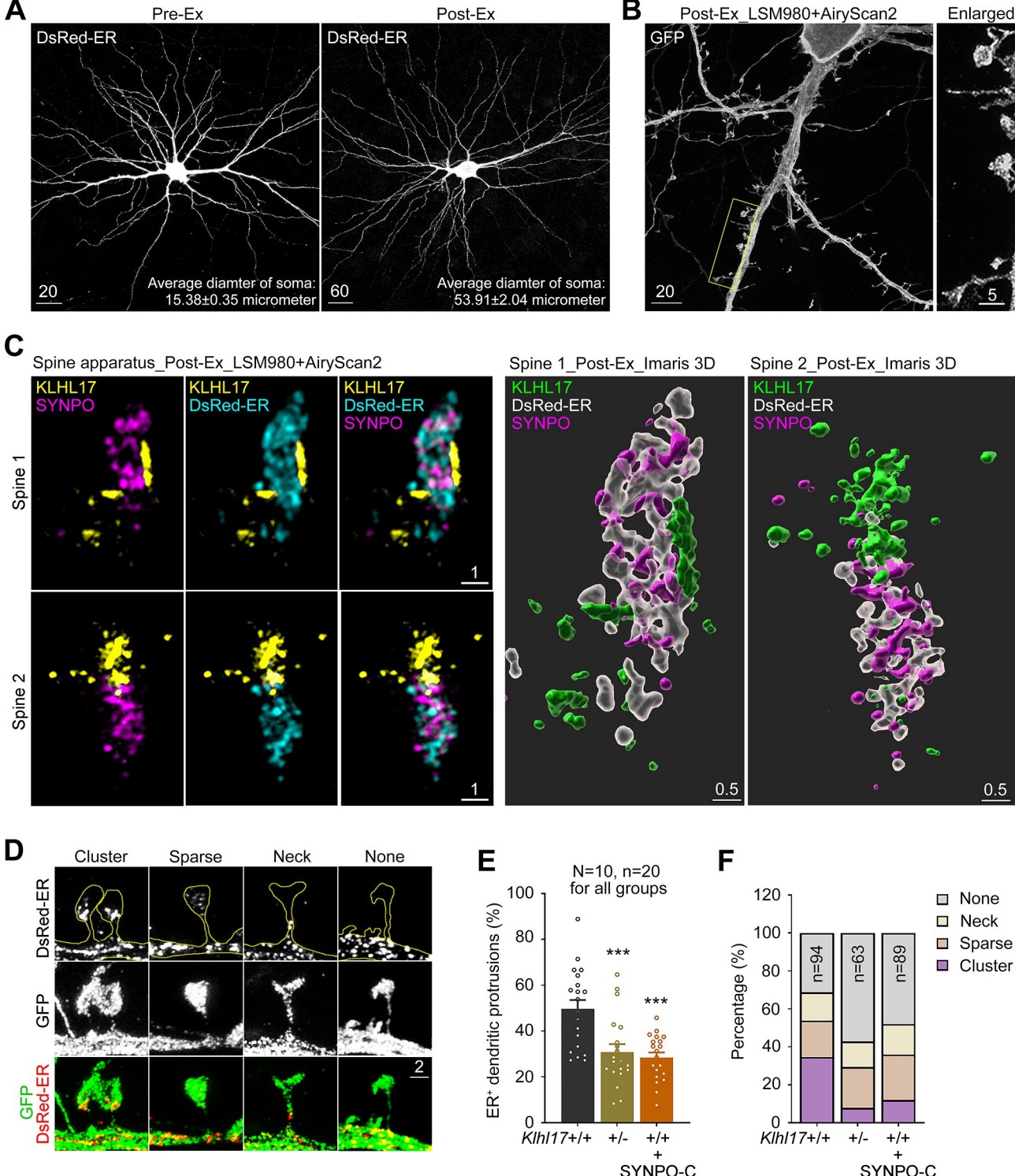

**Fig 10. Expansion microscopy reveals the ultrastructure of the spine apparatus and the function of KLHL17 in ER clustering at dendritic spines.**
Cultured hippocampal neurons were transfected with the indicated plasmids at 12 DIV and harvested for expansion microscopy at 18 DIV. (**A**) Comparison of neuronal morphology before and after expansion (Ex), i.e., pre-Ex vs. post-Ex. The average diameters of somata before and after expansion are indicated. Ten neurons for each group were analyzed. The scale bars of the pre-Ex and post-Ex images are 20 and 60 μm, respectively, which are also labeled. (**B**) Z-series projection of post-expanded GFP-expressing neurons. The image was acquired using an LSM980 laser scanner with AiryScan2. The inset is enlarged and shown at right. Scale bars are 20 and 5 μm, as indicated. (**C**) Left: The distribution patterns of ER (cyan), SYNPO (magenta), and KLHL17 (yellow) in dendritic spines. Two representative dendritic spines are shown. The snapshots of maximal intensity 3D images

containing a Z-series of 2-μm thinkness are shown. Scale bar: 1 μm. Right: a 3D model processed by Imaris are also presented for each spine. ER, white, transparent; SYNPO, magenta; KLHL17, green. Scale bar: 0.5 μm. (**D**) Four distinct distribution patterns of synaptic ER. Neurons were outlined using GFP, whereas ER was labeled by DsRED-ER. Scale bar: 2 μm. (**E**) *Klhl17* deficiency and SYNPO-C expression reduce the synaptic distribution of ER. (**F**) *Klhl17* deficiency and SYNPO-C expression reduce ER clustering at mushroom-like dendritic spines. All data were collected from more than 3 independent experiments. The sample size "*N*" indicates the number of examined neurons and "*n*" represents the number of examined dendritic segments (**E**) or examined dendritic spines (**F**). The sample size is labeled only in 1 panel for the same set of experiments. The data represent mean ± SEM. Individual data points are also shown. *** $P < 0.001$. One-way ANOVA (**E**). The numerical value data and statistical results are available in S1 and S2 Data, respectively. DIV, day in vitro; ER, endoplasmic reticulum; KLHL17, Kelch-like protein 17.

that transcriptional regulation is recognized as a key process impaired by ASD [2,42] and RNA stability has been strongly linked to ASD etiology [43]. Nevertheless, since KLHL17 protein levels are increased upon neuronal activation, a mechanism that enhances translation and/or protein stability likely plays a deterministic role in regulating KLHL17 protein expression. Indeed, many autism-linked risk genes have been shown to control translation [44,45]. NMDAR and its downstream effectors activate the mTOR pathway to enhance translation and up-regulate protein synthesis [46]. Our data also indicate an involvement of NMDAR and mTOR-dependent protein synthesis in regulating KLHL17 protein levels (**Fig 1**). It will be intriguing to elucidate the detailed regulatory mechanism underlying KLHL17 protein expression in the future.

Synaptic plasticity and homeostasis are highly relevant to ASD [6,47,48]. Synaptic upscaling and downscaling both impact neuronal connectivity and influence behaviors [6]. Our study provides evidence that KLHL17 and SYNPO work together to control the distribution of the spine apparatus, thereby impacting calcium dynamics to regulate activity-dependent dendritic spine enlargement. As a regulator of synaptic plasticity, SYNPO forms bigger clusters in neurons in response to synaptic stimulation to potentiate synaptic strength. *Klhl17* deficiency impairs this enlargement of SYNPO clusters upon synaptic stimulation, strengthening the evidence for a role for KLHL17 in controlling synaptic plasticity. However, it remains unclear how KLHL17 regulates SYNPO clustering. Since both KLHL17 and SYNPO are F-actin-binding proteins, F-actin cytoskeletons likely mediate the interaction between both proteins. This speculation is also consistent with our observation that although KLHL17 and SYNPO are functionally linked, the interaction between these 2 proteins is likely indirect for 2 reasons. First, the co-immunoprecipitation efficiency between these 2 proteins proved to be relatively weak. Second, our super-resolution imaging using expansion microscopy also indicated that KLHL17 is adjacent to the SYNPO/ER complex but the signal did not extensively overlap with that of SYNPO or ER. Thus, further experiments are needed to elucidate how KLHL17 associates with SYNPO and whether F-actin cytoskeletons are indeed involved in the association between KLHL17 and SYNPO.

Using the truncated SYNPO-C and SYNPO-N proteins, we found that overexpression of the SYNPO-C fragment, but not the SYNPO-N construct, impaired calcium dynamics in terms of frequency, activity-dependent dendritic spine enlargement and neuronal activation (as indicated by ERK phosphorylation). These phenotypes are identical to those caused by *Klhl17* deficiency. Moreover, SYNPO overexpression rescued all kinds of phenotypes displayed by *Klhl17*-deficient neurons. This evidence supports that KLHL17 and SYNPO work together to control the structural plasticity of dendritic spines. Nevertheless, we also noticed that similar to SYNPO-C, SYNPO-N overexpression also reduced the amplitude of calcium events, yet dendritic spine enlargement and neuronal activation were not affected by this latter, implying that SYNPO may exert another function to control calcium dynamics via its N-terminal region. Moreover, although both *Klhl17* deficiency and SYNPO-C expression narrowed the width of spine heads, *Klhl17* deficiency and SYNPO-C expression actually exhibited

opposing effects on the amplitude of calcium events; *Klhl17* deficiency enhanced the amplitude, whereas SYNPO-C expression reduced it. Thus, the changes in amplitude induced by *Klhl17* deficiency and SYNPO-C expression are unlikely relevant to the reduction of spine volume. Therefore, it would be intriguing to explore other potential molecules or mechanisms responsible for this effect in the future.

Although KLHL17 has been identified previously as a substrate adaptor of CUL3 to mediate degradation of GLUR6 [17], a kainite receptor, it seems unlikely that CUL3-dependent protein degradation plays a role in KLHL17-dependent dendritic spine enlargement. Expression levels of both KLHL17 and GLUR6 are increased as neurons mature [49,50]. If GLUR6 degradation via the CUL3-KLHL17 complex is critical, a gradual reduction in GLUR6 expression during development would be more likely because KLHL17 protein levels are concomitantly increased. Moreover, since GLUR6 degradation is actually enhanced by neuronal activation via CUL3-KLHL17 E3 ubiquitination [17], GLUR6 is also unlikely to mediate KLHL17's activity in controlling activity-dependent dendritic spine enlargement. Therefore, we postulate that KLHL17 exhibits another function, which is to control synaptic calcium dynamics through the spine apparatus and ultimately regulate synaptic plasticity.

## Conclusions

Apart from the involvement of CUL3-dependent protein degradation reported previously [17], we demonstrate another critical function of KLHL17 in regulating synaptic plasticity. We show that mouse homologs of KLHL17 and SYNPO, 2 autism-linked proteins, associate with each other and control the synaptic distribution of the spine apparatus. Given that KLHL17, but not SYNPO, is a brain-specific protein, KLHL17 may guide SYNPO to achieve neuron-specific functions in regulating calcium dynamics at dendritic spines, activity-dependent enlargement of dendritic spines, and neuronal activation (**S4 Fig**). Since *KLHL17* has been linked to neurological disorders, including autism and infantile spasms, our study provides important insights into the functions of KLHL17 in neurons, representing a critical basis for discovering the etiology of KLHL17-related disorders.

## Methods

### Ethics statement

All animal experiments were performed with the approval of the Academia Sinica Institutional Animal Care and Utilization Committee (Protocol No. 18-10-1234) and in strict accordance with its guidelines and those of the Council of Agriculture Guidebook for the Care and Use of Laboratory Animals. This study did not involve human subjects.

### Experimental design

Cultured neurons prepared from *Klhl17*[+/+], *Klhl17*[+/−], and *Klhl17*[−/−] mice were used to investigate how KLHL17 controls structural plasticity. Cortical and hippocampal mixed cultures were used for immunoblotting analysis. Hippocampal cultures were applied in immunofluorescence staining experiments. To activate neuronal cultures, bicuculline was applied for 15 min (to analyze ERK phosphorylation), 2 h (for C-FOS staining), or 6 h (to analyze KLHL17 expression). To study activity-dependent synaptic plasticity, bicuculline treatment was applied for 15 min incubation followed by 30 min recovery. The cultures were then subjected to various experiments, such as immunostaining to monitor dendritic spine density and morphology (based on GFP or Cerulean signals), protein distribution in dendritic spines, and neuronal activation (as indicated by ERK phosphorylation and C-FOS staining), co-immunoprecipitation to investigate

protein–protein interactions between KLHL17 and SYNPO, immunoblotting to examine protein expression levels, and calcium imaging to monitor calcium dynamics.

## Animals and housing

The *Klhl17* mutant mouse lines have been describedin detail previously [13]. All animal experiments were performed with the approval of the Academia Sinica Institutional Animal Care and Utilization Committee (Protocol #18-10-1234) and in strict accordance with the guidelines of the Council of Agriculture Guidebook for the Care and Use of Laboratory Animals. Pregnant mice were housed individually and killed by means of $CO_2$ inhalation. Fetal pups of both sexes were killed by decapitation for neuronal cultures. Mice were housed in the animal facility of the Institute of Molecular Biology, Academia Sinica, under controlled temperature, humidity, and a 12 h light/dark cycle (light-dark: 7 AM to 7 PM) with free access to water and chow (#5010, LabDiet). All efforts were made to minimize animal suffering and to reduce the number of animals utilized.

## Antibodies and reagents

The rabbit polyclonal KLHL17 antibody has been described previously [13]. Rabbit polyclonal HSP90 was a kind gift from Dr. Chung Wang [51]. Commercially available antibodies and chemicals used in this report are as follows: mouse monoclonal Synaptopodin (G1D4, BM5086P, 0.5 µg/ml, ORIGENE); mouse monoclonal beta-actin (AC-74, mouse, 1:5,000, Sigma); mouse monoclonal Myc (9B11, #2276, 1:1,000, Cell Signaling Technology); rabbit polyclonal HA (C29F4, #3724, 1:2,000, Cell Signaling Technology); rat monoclonal HA (3F10, 1:1,000, Roche); chicken polyclonal GFP (ab13970, 1:5,000, Abcam); rabbit polyclonal DsRed (632496, 1:1,000, TAKARA); rabbit polyclonal phosho-ERK (#9101, 1:500, Cell Signaling Technology); rabbit monoclonal C-FOS (9F6, #2250, 1:1,000, Cell Signaling Technology); mouse monoclonal MAP2 (M4403, 1:1,000, Sigma); Alexa Fluor 488-, 555-, and 647-conjugated secondary antibodies (Invitrogen); Tetrodotoxin (TTX; Tocris Bioscience); Bicuculline (Bicu; Tocris Bioscience); 2-amino-5-phosphonopentanoic acid (AP5; Sigma); 2,3-dihydroxy-6-nitro-7-sulfamoyl benzo(f)quinoxaline (NBQX; Tocris Bioscience); NMDA (Tocris Bioscience); Cyclohexamide (CHX, Sigma); Ryanodine (Tocris); 6-((acryloyl)amino)hexanoic acid (Acryloyl-X, SE, ThermoFisher, A-20770); Sodium acrylate (Sigma-Aldrich, 408220); Acrylamide (Sigma-Aldrich, A9099); N, N'-methylenebisacrylamide, (Thermo Fisher, 1551624); Ammonium persulfate, (APS, Sigma-Aldrich, A9164); N,N,N,,N,-Tetramethylethylenediamine (TEMED, Sigma-Aldrich, T7024); 4-Hydroxy-TEMPO (ALDRICH, 176141); Sodium chloride (J.T.Baker, 3624–69); Trypsin-EDTA solution, 0.05% (Gibco, 25300–054).

## Plasmid construction

Myc-tagged KLHL17 expression plasmid and N-terminal (KLHL17-N) and C-terminal (KLHL17-C) truncated mutants have been described previously [13]. To generate HA-tagged SYNPO expression plasmids, full-length SYNPO (residue 1–690), N-terminal SYNPO-N (residues 1–299), and C-terminal SYNPO-C (residues 300–690) were PCR-amplified from GFP-synaptopodin (a kind gift from Dr. Marina Mikhaylova) and subcloned into vector GW1-HA digested by *XmaI*. pCAG-GFP (#11150, Addgene) and pEGFP-actin (#6616–1, Clontech) were used to outline cell morphology. pCAG-DsRed-ER, described previously [52], was used to monitor the ER cell distribution. To construct pCAG-Cerulean, the GFP fragment of the pCAG-GFP vector was removed by *EcoRI* and *BglII* digestion. The Cerulean fragment was released from pcDNA3.1/Zeo(-)::Cerulean (a kind gift from Dr. Chin-Yin Tai) by means of *BamHI* and *EcoRI* digestion and then subcloned into linearized pCAG vector. For

pCAG-GCaMP6S-P2A-nls-dTomato, the GFP fragment of the pCAG-GFP vector was removed by *EcoRI* and *NotI* digestion, followed by Klenow fill-in. The GCaMP6s-P2A-nls-dTomato fragment was released from AAV-EF1a-DIO-GCaMP6S-P2A-nls-dTomato (#51082, Addgene) by means of *BamHI* and *EcoRI* digestion, followed by Klenow fill-in, and then sub-cloned into linearized pCAG vector via blunt-end ligation. Ctrl-miR and *Klhl17*-miR deployed in our *Klhl17* knockdown experiments have been described previously [13].

## Hippocampal neuronal culture, transfection, and neuronal treatment

Cortical and hippocampal neurons from E18.5 mouse embryos were cultured and transfected as described previously [13,25]. To generate *Klhl17*$^{+/+}$, *Klhl17*$^{+/−}$, *and Klhl17*$^{−/−}$embryos for experiments, male *Klhl17*$^{+/−}$mice were mated with female *Klhl17*$^{+/−}$mice. For immunofluorescence analysis, neurons at a density of 500,000 cells per well were seeded onto coverslips (18 mm in diameter and 0.17 to 0.2 mm in thickness) coated with poly-L-lysine. Transfection using calcium phosphate precipitation was performed at 12 DIV and neurons were subjected to immunostaining at 18 DIV. To study the effect of neuronal activity on the KLHL17 expression levels, cultured neurons were treated with TTX (1 μM) or bicuculline (40 μM) for 6 h. To clarify the effect of activity-induced KLHL17 induction, inhibitors of NMDA receptor (AP5, 100 μM), AMPA receptor (NBQX, 100 μM), protein synthesis (CHX, 10 μM), and mTOR (Rapamycin, 10 nM) were added into cultures with bicuculline for 6 h. To study activity-dependent events, cultured neurons were treated with bicuculline for 15 min or 2 h to induce ERK phosphorylation or C-FOS expression, respectively. To examine activity-induced plasticity, cultured neurons were treated with bicuculline for 15 min and then subjected to washout for a further 30-min recovery to monitor spine head enlargement and SYNPO clustering.

## Immunofluorescence analysis

**Cultured hippocampal neurons.** Neurons were washed with phosphate-buffered saline (PBS) and fixed with 4% paraformaldehyde and 4% sucrose in PBS for 10 min, followed by permeabilization with 0.2% Triton X-100 in PBS for 15 min at room temperature. After blocking with 10% bovine serum albumin (BSA), cells were incubated with primary antibodies diluted in PBS containing 3% BSA and 0.1% horse serum at 4˚C overnight. After washing with PBS several times, cells were incubated with secondary antibodies conjugated with Alexa Fluor-488, -555, and -647 (Invitrogen) for 2 h at room temperature.

**Hippocampal CA1 neurons in mouse brains.** To visualize dendritic spines in mouse brains, Thy1-YFP mice were crossed with *Klhl17* mutant mice. YFP signals were used to outline neuronal morphology. After perfusion with 4% paraformaldehyde in PBS, the brains were dissected and postfixed with 4% paraformaldehyde overnight at 4˚C with gentle shaking. Fifty-μm-thick brain sections were cut using a vibratome (VT1200S, Leica). Brain slices were permeabilized with 0.3% TritonX-100 in TBS for 15 min and blocked with TNB buffer (0.5% blocking reagent in TBS, #FP1012, PerkinElmer) for 2 h at room temperature. The primary anti-synaptopodin antibody (0.2 μg/ml) was then added for 2 days at 4˚C. After washing, brain slices were incubated with Alexa fluor 555-conjugated secondary antibodies for 2 h at room temperature. Details of the protocol for CA1 analysis have been described previously [53].

**Imaging conditions and processing.** The Z-series images (1 μm/single stack) were acquired at room temperature using a confocal microscope (LSM700; Carl Zeiss) equipped with a 63×/NA 1.4 oil objective lens and Zen 2009 software (Carl Zeiss) and analyzed using ImageJ/FIJI software (1.8v, NIH). Super-resolution images were acquired using an LSM980 system with Airyscan2 (alpha Plan-Apochromat 100×/1.46 Oil objective). The 3D reconstruction images were processed in Imaris (8.3v, Bitplane), and modeling was based on fluorescence

signals. For quantitation, the same set of experimental samples was acquired under identical confocal microscopy settings. Post-acquisition adjustment was avoided.

**Quantification.** For each set of experiments, we routinely collected neuronal images from at least 2 independent preparations in a blind fashion. The density and size of dendritic spines vary among different types of neurons. Based on their morphological features, we noticed that median-sized neurons with a soma diameter of approximately 15 to 18 μm and 3 to 8 primary dendrites represent the major population in the transfected hippocampal culture. Therefore, we selected this group of neurons for further analysis. For each neuron, 3 clearly recognizable dendrites of comparable thickness were chosen to quantify dendritic spine density and size. Given that the distance from the soma to the dendritic spine also influences the density and size of dendritic spines, we further specifically analyzed 20-μm dendritic segments that were 20 μm apart from the soma. The spine number of those 20-μm dendritic segments was determined as dendritic spine density. All dendritic spines on the selected dendritic segments were subjected to width and length analysis. In doing so, we minimized in-group variation as well as personal bias. In the respective figures, the sample size "$N$" indicates the number of examined neurons and "$n$" represents the number of examined dendrites or dendritic spines.

## Expansion microscopy

Immunostained neurons, prepared as described above, were subjected to expansion microscopy as described previously [41], except that tyramide signal amplification was not applied. Briefly, the stained cells were incubated with 0.1 mg/ml Acryloyl-X, SE in PBS overnight at 4°C. After washing 3 times with PBS, the coverslip with stained neurons was placed on acrylamide solution (with the cell-attached side facing down) at 37°C for 2 h. To make the acrylamide hydrogel, 2 μl 10% TEMED, 1 μl 1% TEMPO, 1 μl water, and 2 μl 10% APS were freshly added into 188 μl acrylamide monomer solution (8.6% sodium acrylate, 2.5% acrylamide, 0.15% N,N'-methylenebisacrylamide, and 11.7% sodium chloride in PBS). The solution was sufficient to make a 0.5-mm-thick hydrogel on a coverslip with a diameter of 15 to 18 mm. After gelation was completed, the hydrogel was immersed in trypsin digestion solution (1 volume of trypsin-EDTA solution and 9 volumes of low salt digestion buffer containing 50 mM Tris (pH 8.0), 1 mM EDTA, 0.5% Triton-X-100) and incubated at room temperature for at least 20 h. The digested hydrogel was then gently transferred to a 10-cm dish filled with excess Milli-Q water for free expansion for at least 2 h, with at least 3 water changes. The expanded samples were then imaged at room temperature using an LSM980 laser scanning confocal microscope equipped with Airyscan 2 multiplex mode SR or SR-4Y (2× sampling). An LD LCI Plan-Apochromate 40×/1.2 Imm Corr DIC M27 objective was used for Z-series image acquisition with an interval of 0.2 μm. The Z-series images were further processed for 3D surface construction using Imaris software (v10.0.1).

The correlation and colocalization were analyzed by ImageJ/FIJI software with JACoP plugin tool.

## Transfection, immunoblotting, and immunoprecipitation of Neuro-2A cells

Mouse neuroblastoma Neuro-2A (N2A) cells (ATCC CCL-131) were transfected with the indicated plasmids using PolyJet (SL100688, SignaGen Laboratories) according to the manufacturer's instructions. For immunoprecipitation analysis, 1 day after transfection, transfected N2A cells were washed with PBS and solubilized in lysis buffer (PBS (pH 7.4), 1% Triton X-100, 2 mM PMSF, 2 μg/ml aprotinin, 2 μg/ml leupeptin, 2 μg/ml pepstatin, and 10 μm MG132) at 4°C for 30 min. The lysates were centrifuged at 15,000×$g$ for 30 min to remove the cell debris.

The solubilized extract was subjected to immunoprecipitation using Myc or HA antibodies combined with protein A sepharose. After mixing by rotation at 4°C overnight, the precipitates were sequentially washed twice with each of the following buffers: 1% Triton X-100/PBS, 0.5% NP-40/0.5 M LiCl/50 mM Tris (pH 8), 0.5 M LiCl/50 mM Tris (pH 8), 10 mM Tris (pH 8). The precipitated proteins were then separated using SDS-PAGE and analyzed by immunoblotting. The immunoblotting results were recorded using an ImageQuant LAS 4000 system with ImageQuant LAS 4000 Biomolecular Imager software (GE Healthcare) and quantified using ImageJ/FIJI software. Relative expression levels of KLHL17 were normalized against HSP90 or actin. The uncropped full-size blots are available in **S1 Raw Images**.

## Quantification of the distribution of synaptic proteins

To examine the distribution of synaptic proteins along dendritic protrusions, line scanning using ImageJ/FIJI software was performed. A 0.5-μm wide line starting from the top of the dendritic protrusion and ending at the dendritic shaft was drawn to quantify protein intensities. The line chart represents the location of synaptic proteins within the dendritic protrusions. Protrusions with a signal intensity of >20 units were defined as immunoreactivity-positive.

## RNA extraction and quantitative real time-PCR (RT-PCR)

To detect the expression levels of *Klhl17* RNA through development, RNA was isolated from cultured neurons and mouse whole brain at the indicated time points (1, 7, 14, 21 DIV or postnatal days 1, 7, 14, 21, respectively). Total RNA was extracted using TriPure reagent according to the manufacturer's instructions (Roche), followed by DNase I (New England BioLabs) digestion for 30 min at 37°C to remove contaminating DNA. Reverse transcription was performed using a Transcriptor First Strand cDNA Synthesis kit (Roche) with an oligo(dT)18 primer. Quantitative RT-PCR analysis was performed using a LightCycler480 (Roche) and Universal Probe Library probes (UPL; Roche) system. The primers and their paired probes were designed using the Assay Design Center Web Service (Roche). Primer sets and probe numbers are: *Klhl17* (F: AGGCAGCATGTACCAAGGTT and R: AGGAGGAAGTCTCGG CTCA, probe #34) and *Hprt* (F: CTTCCTCCTCAGACCGCTTT and R: GGTTCATCATC GCTAATCACG, probe #95). Relative expression levels of *Klhl17* were normalized against the levels of *Hprt* measured at the same time on the same reaction plate. Samples were assayed in triplicate and then averaged to represent the data of a single experiment.

## Calcium recording

Cultured neurons were transfected with the calcium indicator GCaMP6s at 12 DIV and spontaneous calcium events were recorded by monitoring GCaMP6s fluorescence signals at 18 to 19 DIV. Neurons were transferred onto a recording chamber and incubated in a regular maintaining medium plus extra $CaCl_2$ (2 mM). Calcium events were monitored according to the fluorescence intensity of GCaMP6s ($\lambda$ex: 488 nm) with an inverted Spinning Disk confocal microscope equipped with a 100× objective lens (CFI Apochromat TIRF 100XH/NA 1.49), CCD (Andor iXON Ultra 888), and Metamorph (7.8v) acquisition software. The scanning area was $25 \times 25 \times 1$ μm and the scanning speed was approximately 5 frames per second. Relative fluorescence units (RFUs) of GCMP6s signals in individual dendritic protrusions were quantified using the Plot Z-axis Profile tool (ImageJ/FIJI software) and the data was used to determine ΔF/F. We set the fluorescence intensity of the basal level as 1. The relative amplitude of each peak was then determined. Only peaks with a relative amplitude >1.5 were considered

significant for analysis. The frequency, interval time, and amplitude of calcium events were then determined. Statistical analyses were performed using Prism9 software (Graphpad).

## Quantification and statistical analysis

Cultured neurons were randomly assigned to different treatments. To avoid potential personal bias, the data were collected and analyzed blindly by having other lab members relabel the samples. Image measurements, including morphometry analysis, neuronal activation, synaptic distribution, and immunoblotting analysis, were carried out using ImageJ/FIJI software. To analyze activity-dependent calcium events, a series of acquired images were quantified using ImageJ/FIJI software. Two-group comparisons were analyzed with an unpaired and two-tailed Student's $t$ test. For more than 2 group comparisons, a one-way analysis of variance (ANOVA) with Bonferroni's multiple comparison post hoc test was performed. For morphometry of dendritic spines and SYNPO rescue experiments, the data were compared by two-way ANOVA with Bonferroni's multiple comparison post hoc test. All of the results shown in the cumulative probability distribution were analyzed using a Kolmogorov–Smirnov test (https://www.aatbio.com/tools/kolmogorov-smirnov-k-s-test-calculator). Statistical analysis and graphical outputs were conducted in Prism 9 software (Graphpad). For all comparisons, $P < 0.05$ was considered significant. All original data used for analyses are summarized in **S1 Data**. The complete results of statistical methods and analyses for this study are available in **S2 Data**.

## Supporting information

**S1 Fig. Bicuculline treatment enhances ERK phosphorylation and C-FOS expression.** Wild-type neurons at 18 DIV were treated with bicuculline (40 μM) for 15 min to investigate ERK phosphorylation (**A**) or for 2 h to assess expression of C-FOS (**B**). The levels of ERK phosphorylation and C-FOS expression are very low in the absence of bicuculline under our experimental conditions. Scale bars: (**A**), (**B**) 50 μm.
(TIF)

**S2 Fig. Enlarged images of Fig 6A and 6C.** (**A**) Enlarged image of the left panel in Fig 6A, which was acquired using a LSM700 system. (**B**) Enlarged image of the left panel in Fig 6C, which was acquired using an LSM980 system with AiryScan2. Scale bar: (**A**) 20 μm; (**B**) 5 μm.
(TIF)

**S3 Fig. Colocalization coefficients of KLHL17, SYNPO, and ER revealed by expansion microscopy.** The correlation and colocalization of KLHL17, SYNPO, and ER in the spine apparatuses shown in Fig 10C were analyzed using ImageJ/FIJI. (**A**) Spine 1. (**B**) Spine 2. Upper panel, Pearson's correlation; lower panel, colocalization percentage. X-axis: section number of Z-series. The numerical value data are available in S1 Data.
(TIF)

**S4 Fig. KLHL17 associates with SYNPO to control the synaptic distribution of the spine apparatus, calcium dynamics, neuronal activation, and plasticity. (A)** KLHL17 associates with SYNPO, a marker of the spine apparatus, to control the synaptic ER distribution, thereby influencing calcium dynamics at dendritic spines and consequently regulating activity-dependent events. **(B)** Comparison of *Klhl17*-deficient mice and wild-type mice.
(TIF)

**S1 Raw Images. Uncropped immunoblots used in the current study.**
(PDF)

**S1 Data. Raw data underlying Figs 1–10.**
(XLSX)

**S2 Data. All statistical methods and results.**
(XLSX)

## Acknowledgments

We thank Dr. Marina Mikhaylova at Zentrum für Molekulare Neurobiologie Hamburg for SYNPO plasmid and discussion, the Imaging Core and Animal Facility of the Institute of Molecular Biology, Academia Sinica, for excellent technical assistance, and Dr. John O'Brien conducted English editing and members of Y.-P.H.'s laboratory relabeled samples for blind experiments.

## Author Contributions

**Conceptualization:** Hsiao-Tang Hu, Yi-Ping Hsueh.

**Data curation:** Hsiao-Tang Hu, Yung-Jui Lin.

**Funding acquisition:** Yi-Ping Hsueh.

**Investigation:** Hsiao-Tang Hu, Yung-Jui Lin, Ueh-Ting Tim Wang, Sue-Ping Lee, Yae-Huei Liou, Bi-Chang Chen.

**Methodology:** Hsiao-Tang Hu, Yung-Jui Lin, Ueh-Ting Tim Wang, Sue-Ping Lee, Yae-Huei Liou, Bi-Chang Chen.

**Project administration:** Yi-Ping Hsueh.

**Supervision:** Yi-Ping Hsueh.

**Visualization:** Hsiao-Tang Hu, Yung-Jui Lin, Sue-Ping Lee, Yae-Huei Liou.

**Writing – original draft:** Hsiao-Tang Hu, Yung-Jui Lin, Yi-Ping Hsueh.

**Writing – review & editing:** Hsiao-Tang Hu, Yung-Jui Lin, Ueh-Ting Tim Wang, Sue-Ping Lee, Yae-Huei Liou, Bi-Chang Chen, Yi-Ping Hsueh.

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
