## [Editor Report · Decision Letter 0]

19 Dec 2022

Dear Yi-Ping, 

Thank you for submitting your manuscript entitled "Autism-related KLHL17 and SYNPO act in concert to control activity-dependent dendritic spine enlargement by regulating the location of the spine apparatus and calcium dynamics to impact mouse neural activation" for consideration as a Research Article by PLOS Biology.

Your manuscript has now been evaluated by me in consultation with the rest of the PLOS Biology editorial staff and I am writing to let you know that we would like to send your submission out for external peer review. Please note that, due to the upcoming holidays, we have not been able to get feedback from an Academic Editor with relevant expertise. We are therefore not yet making any formal call yet on whether the overall advance is appropriate for PLOS Biology or not and will be looking for reviewer support on this point.

Before we can send your manuscript to reviewers, we need you to complete your submission by providing the metadata that is required for full assessment. To this end, please login to Editorial Manager where you will find the paper in the 'Submissions Needing Revisions' folder on your homepage. Please click 'Revise Submission' from the Action Links and complete all additional questions in the submission questionnaire. During this process you will be given the opportunity to suggest and exclude potential reviewers. We do find it helpful when authors provide suggestions for reviewers who have the appropriate expertise to cover the various angles in their submission, and who are without conflicts of interest. There are no limits on the number of people you may suggest. We limit the number of excluded reviewers to 3 individuals. 

Once your full submission is complete, your paper will undergo a series of checks in preparation for peer review. Given the upcoming holidays and office closures, there may be some delay in this process. I am also currently on holidays until Jan 3rd, 2023, so will not be able to start lining up reviewers on this work until the first week of Jan 2023. I apologize in advance for these delays.

To provide the metadata for your submission, please Login to Editorial Manager (https://www.editorialmanager.com/pbiology) within 9 working days, i.e. by Dec 30th.

Kind regards,

Kris

Kris Dickson, Ph.D., (she/her)

Neurosciences Senior Editor/Section Manager

PLOS Biology

kdickson@plos.org

---

## [Decision Letter · Decision Letter 1]

10 Feb 2023

Dear Dr Hsueh,

Thank you for your patience while your manuscript "Autism-related KLHL17 and SYNPO act in concert to control activity-dependent dendritic spine enlargement by regulating the location of the spine apparatus and calcium dynamics to impact mouse neural activation" was peer-reviewed at PLOS Biology. It has now been evaluated by the PLOS Biology editors, an Academic Editor with relevant expertise, and by several independent reviewers. 

In light of the reviews, which you will find at the end of this email, we would like to invite you to revise the work to thoroughly address the reviewers' reports.

Given the extent of revision needed, we cannot make a decision about publication until we have seen the revised manuscript and your response to the reviewers' comments. Your revised manuscript is likely to be sent for further evaluation by all or a subset of the reviewers.

**IMPORTANT - SUBMITTING YOUR REVISION**

*Re-submission Checklist*

*Published Peer Review*

*PLOS Data Policy*

*Blot and Gel Data Policy*

Sincerely,

Kris

Kris Dickson, Ph.D., (she/her)

Neurosciences Senior Editor/Section Manager

PLOS Biology

kdickson@plos.org

REVIEWS:

>Do you want your identity to be public for this peer review?

Reviewer #1: No

Reviewer #2: No

Reviewer #3: No

Reviewer #1: Hu et al. study the importance of two postsynaptic proteins ,KLHL17 (actinfilin) and synaptopodin, on the structure and function of dendritic spines. They find that KLHL17 is upregulated during development by neuronal activity (NMDAR-dependent). While in wild-type neurons high activity leads to enlarged spine heads, this is not the case in KLHL17 knock-down or knock-out neurons. KLHL17 deficiency reduces ERK and FOS expression after high activity and reduces the frequency of spontaneous calcium events in spines. Fewer spines contain endoplasmic reticulum. As they show with conventional confocal microscopy and Airyscan, KLHL17 colocalizes tightly with synaptopodin in dendritic spines, suggesting it is part of the spine apparatus. Indeed, KLHL17 deficiency or KO reduces the fraction of spines containing a spine apparatus, and synaptopodin overexpression rescues the deficits of KLHL17-deficient neurons. Overexpression of truncated forms of synaptopodin suggests that binding of KLHL17 to synaptopodin (perhaps indirect) is needed for efficient spine apparatus formation and spine head enlargement.

The study is easy to read and understand due to its clear structure and logic. The data quality is excellent, statistical tests are appropriate and all controls are in place. The role of KLHL17 in spine apparatus assembly and spine head structure is clearly demonstrated, providing a possible mechanistic explanation for its synaptic function and association with autism. I have only minor suggestions for improvements.

Minor points:

1. Figure 4B and E: Please provide relative change in fluorescence (dF/F) instead of relative fluorescence units (RFU) as a measure of calcium concentration changes. Although spine calcium transients were analyzed in spine heads, the example (A) shows a dendritic calcium wave. Could you provide a more detailed analysis of dendritic vs. spine head Ca signals? For example, showing in (B) the fluorescence time course in both spine head and parent dendrite. Ideally, it would be nice to analyze the amplitude of spine Ca transients that are NOT associated with dendritic Ca events. Right now, the differences could be explained by lower levels of spontaneous activity in the +/- cultures. The open question is: Does KLHL17 deficiency (=lack of spine ER) directly affect spine calcium handling? Perhaps a discussion point: The higher Ca amplitudes could be due to the smaller volume of spines in KLHL17+/- neurons (less dilution). 

2. Fig. 5D: SPH+ ("spine head-positive") is a very unusual term and abbreviation, I had to search to find an explanation in the text. Would this correspond to the frequently used categories 'mushroom' and 'thin'? Please find a better label for the columns, even if it needs a bit more space. 

3. Page 12: "cytoskelens"

4. Title: Too long, the first half is sufficient. 

Reviewer #2: This work from Hu and colleagues explores the roles of KLHL17, a protein which mutation is associated with autism spectrum disorder. The authors identify a possible mechanism for the regulation of activity-dependent dendritic spines enlargement mediated by KLHL17. They suggest that the associated action of KLHL17 and synaptopodin promotes the insertion of the spine apparatus (ER) in dendritic spines and this effect is responsible for the dendritic spine enlargement, possibly thanks to the Calcium released by the ER in this compartment.

The experiments are of excellent quality and the story is of great interest for the cellular neuroscience field.

However, there are some aspects that need clarifying.

Major points:

I feel like the evidence that KLHL17 regulates dendritic spine enlargement via SYNPO and via the insertion of the spine apparatus in dendritic spines is somehow still unclear, especially considering the importance of actin polymerisation in this phenomenon and the already demonstrated role of KLHL17 on actin polymerisation. It is also possible that KLHL17 regulates actin polymerisation, causing smaller dendritic spines which are not able to accommodate ER, without a direct action of KLHL17 on the ER. One experiment would be required:

- does reduction of SYNPO levels in a KLHL17 -/- background further affect spine apparatus insertion and activity dependent spine enlargement? This would clarify if they participate together in this process or they have independent actions on the same process.

1) Figure 4: that the changes in frequency and amplitude of calcium events upon KLHL17 reduction are dependent on a reduced efflux of calcium from the ER is not demonstrated. The same experiment should be performed with NMDA-R blockers (is the defect still visible) and separately with blockers of ER calcium channels

2) Can the author perform calcium imaging experiments with and without KLHL17 upon the activity paradigm implemented in all other experiments (15 min Bicuculline + recovery)?

3) The ER or near ER localisation of KLHL17 is somehow not fully convincing. The airyscan2 imaging used in Figure 6C is not a full super-resolution approach with a nominal resolution of 90nm (and realistically the resolution will be lower). Would it be possible for the authors to perform STORM or STED experiments for this? Also imaging the endogenous KLHL17 and SYNPO would be important. 

4) I do not think that showing statistical analysis per "dendritic segment" or per "spine" is acceptable. At least the analysis should be shown per neuron averaging the dendritic segments or the spines belonging to the same neurons. 

5) All bar graphs should show individual data points.

Minor

Figure 1F; GFP/Myc-KLHL17 label is unclear of what is shown in the image. I think it should say myc-KLHL17 only.

Figure 1H; the intensity of myc-KLHL17 in the image in the panel Bicu+CHX is still higher than ctrl, and as such not representative of what is shown in the bar graph in panel I.

Figure 3: Can the authors please add pERK and c-Fos images at baseline condition? 

Figure 3: the DAPI signal is not really visible when the figure is printed, can the author increase brightness?

Figure 3: Can the authors show pERK also via western blotting together with total ERK intensity? This is important because it would allow to show pERK/ERK signal.

Figure 5A. Can the authors please show individual colour panels?

Figure 5E: the GFP outline is not visible when the figure is printed. Can the authors make it brighter or thicker?

Figure 6F, I: the GFP outline is not visible when the figure is printed. Can the authors make it brighter or thicker?

Figure 6A and C: the image is very dim when printed, can the authors make it brighter.

Fig 7I: as before, can the authors measure pERK and ERK via western blot and show pERK/ERK ratio?

Figure 8 B: coIPed HA is barely visible, can the author show a longer exposure image?

Figure 8D-F: it would be interesting to see what effect the expression of SYNPO full length has on the percentage of dendritic spines positive for ER? 

Figure 8D-F: it would be important to measure the expression levels of Synpo-N and Synpo-C to verify they are similar.

Figure 9I: as before, can the authors measure pERK and ERK via western blot and show pERK/ERK ratio?

Reviewer #3: In this very interesting manuscript the authors report the discovery and characterization, as well as the functional impact, of the interaction between two dendritic spine proteins, KLHL17 and SYNPO. Dendritic spine plasticity, as well as these genes, are involved in neurodevelopmental disorders such as autism, hence the work has both basic and translational relevance. The experiments are well-designed, proper controls and statistics are used, and the experiments are performed at high standards of quality. The logical flow is correct. The major findings are that KLHL17 expression is regulated by neuronal activity, and it interacts directly with synaptopodin, a protein associated with ER in dendritic spines. They then characterize this interaction, and its effects on ER in spines, neuronal activity, dendritic spine morphology, and calcium signaling. The major novelty in my opinion are the interaction of the two proteins, the regulation of synaptic ER by their interactions, and the methodological use of superresolution microscopy. However, these aspects would need to be developed in more depth for the study to reach its full potential. I suggest the following major revisions:

- in Figure 5, superresolution images are shown as examples, but not employed throughout. Using superresolution to quantify the detailed morphological alterations in dendritic ER would be very interesting

- In Figure 8 as well, superresolution images and an IMARIS 3D reconstruction are shown as examples, but not analyzed extensively. I would suggest to use superresolution throughout and employ detailed quantification of the changes they observe.

- while the interaction between the proteins is examined using biochemical methods, there are powerful imaging tools for the analysis of these interactions within individual cells and spines. I would recommend that such a method be used, to enhance novelty and impact.

---

## [Decision Letter · Decision Letter 2]

30 Jun 2023

Dear Dr Hsueh,

Thank you for your patience while we considered your revised manuscript "Autism-related KLHL17 and SYNPO act in concert to control activity-dependent dendritic spine enlargement and the spine apparatus" for consideration as a Research Article at PLOS Biology. Your revised study has now been evaluated by the PLOS Biology editors, the Academic Editor and the original reviewers.

In light of the reviews, which you will find at the end of this email, we are pleased to offer you the opportunity to address the remaining points from the reviewers in a revision that we anticipate should not take you very long. We will then assess your revised manuscript and your response to the reviewers' comments with our Academic Editor aiming to avoid further rounds of peer-review, although might need to consult with the reviewers, depending on the nature of the revisions.

The Academic Editor notes that additional justification needs to be provide for the per-spine analysis as in this case the variability appears to be on a cell-by-cell bases. If you show data per-spine, additional information needs to be included as noted by Reviewer 3. There is no justification for per-dendritic segment as these segments are part of the same cell, and appear to vary as a group depending from what cell they came. Therefore, this analysis should be re-done on a per-neuron basis.

**IMPORTANT - SUBMITTING YOUR REVISION**

*Resubmission Checklist*

*Published Peer Review*

*PLOS Data Policy*

*Blot and Gel Data Policy*

Kind regards,

Christian

Christian Schnell, PhD

Senior Editor

PLOS Biology

cschnell@plos.org

REVIEWS:

Reviewer #1: The authors have incorporated my suggestions and improved their data analysis. I still have some questions about the calcium imaging data presented in Fig. 4: 

Fig. 4F: I assume the length of the dF/F scale bar is equal to 1 (100% fluorescence change, doubling of absolute intensity). The spine-only transient then has an amplitude of about 0.75. It is not stated whether this example is from WT or Klhl17+/- (please add this information). In Fig. 4I, however, none of the individual data points is below 1 dF/F. What is going on? Please show a representative example from the middle of the distribution. 

Also, in Fig. 4I, please always plot the amplitude in the spine (should be close to zero for dendrite-only events, right?).

Reviewer #2: The authors answered to questions raised by the reviewers.

Reviewer #3: The authors present a well-thought-out study with logical progression from establishing the activity-dependent function of KLHL17 in neurons to describing how its interaction with SYNPO and the ER participates in the regulation of dendritic morphology and calcium dynamics. The importance of the research is underscored by the relevance for human disease as well as the discovery of a novel function for the brain-specific protein KLHL17. The impact of the study is improved as a result of the authors' response to previous review comments with the inclusion of additional experiments to support their conclusions. In particular, the super-resolution images reveal additional structural information which is very relevant. Some minor improvements could be made.

Point 1: Regarding Fig 10, the authors comment that KLHL17 signal is adjacent to SYNPO/ER, as well as intermingled together in complexes. I suggest that they quantify the percentage of single colocalization with either SYNPO or ER, as well as the triple complex.

Point 2: I agree with Reviewer #2's comment regarding the fact that for some experiments it would be more appropriate to do the statistical analysis per neuron instead of per spine, for example for the calcium imaging in Fig. 4A-E, Fig. 7A-H, Fig. 9A-D. However, I think that the per spine analysis can still be accepted provided that the authors analyze a similar number of spines for each individual neuron and that they mention the number of neurons included per condition in the figure legends.

Point 3: The legend for S1 figure does not match the panels. Please correct.

Point 4: Page 3 "actifilin" spelling

---

## [Editor Report · Decision Letter 3]

20 Jul 2023

Dear Yi-Ping,

Thank you for your patience while we considered your revised manuscript "Autism-related KLHL17 and SYNPO act in concert to control activity-dependent dendritic spine enlargement and the spine apparatus" for publication as a Research Article at PLOS Biology. This revised version of your manuscript has been evaluated by the PLOS Biology editors and the Academic Editor.

Based on the reviews on our Academic Editor's assessment of your revision, we are likely to accept this manuscript for publication, after you have addressed the following data and other policy-related requests.

- In the "Financial disclosure" section in the manuscript details, please also provide the URLs of any funder's website.

DATA POLICY:

Please also ensure that figure legends in your manuscript include information on where the underlying numerical value data can be found (for example S1 Data), and ensure your supplemental data file/s has a legend.

We require the original, uncropped and minimally adjusted images supporting all blot and gel results reported in an article's figures or Supporting Information files. We will require these files before a manuscript can be accepted so please prepare and upload them now. Please carefully read our guidelines for how to prepare and upload this data: https://journals.plos.org/plosbiology/s/figures#loc-blot-and-gel-reporting-requirements

We expect to receive your revised manuscript within two weeks. 

*Published Peer Review History*

*Press*

Sincerely,

Christian

Christian Schnell, PhD

Senior Editor,

cschnell@plos.org,

PLOS Biology

---

## [Editor Report · Decision Letter 4]

24 Jul 2023

Dear Yi-Ping,

Thank you for the submission of your revised Research Article "Autism-related KLHL17 and SYNPO act in concert to control activity-dependent dendritic spine enlargement and the spine apparatus" for publication in PLOS Biology. On behalf of my colleagues and the Academic Editor, Matthew Dalva, I am pleased to say that we can in principle accept your manuscript for publication, provided you address any remaining formatting and reporting issues. These will be detailed in an email you should receive within 2-3 business days from our colleagues in the journal operations team; no action is required from you until then. Please note that we will not be able to formally accept your manuscript and schedule it for publication until you have completed any requested changes.

PRESS

Best wishes,

Christian 

Christian Schnell, PhD, PhD

Senior Editor

PLOS Biology

cschnell@plos.org